# OLD MEMORIES DIE HARD: UNDERSTANDING CHALLENGES OF PRIVACY UNLEARNING IN LARGE LANGUAGE MODELS

## ABSTRACT

Large language models (LLMs) often memorize private information during training, raising serious privacy concerns. While machine unlearning has emerged as a promising solution, its true effectiveness against privacy attacks remains unclear. To address this, we propose PriLeak, a new evaluation framework that systematically assesses unlearning robustness through three-tier attack scenarios: direct retrieval, in-context learning recovery, and fine-tuning restoration; combined with quantitative analysis using forgetting scores, association metrics, and forgetting depth assessment. Our study exposes significant weaknesses in current unlearning methods, revealing two key findings: 1) unlearning exhibits ripple effects across gradient-based associated data, and 2) most methods suffer from shallow forgetting, failing to remove private information distributed across multiple model layers. Building on these findings, we propose two key strategies: association-aware core-set selection that leverages gradient similarity, and multi-layer deep intervention by progressive learning rates and representational constraints. These strategies represent a paradigm shift from shallow forgetting to deep forgetting.

## 1 INTRODUCTION

Large language models (LLMs) inevitably memorize personally identifiable information (PIIs) during training on web-scale data, raising serious privacy concerns when models are publicly deployed (Lukas et al., 2023). This has drawn regulatory attention such as the European General Data Protection Regulation (GDPR) (Politou et al., 2018), which grants individuals the "right to be forgotten." Moreover, the U.S. FTC has established the precedent of Algorithmic Disgorgement, mandating the deletion of not just data but also weights and algorithms trained on non-compliant data (Goland, 2022). Thus, unlearning is a prerequisite for regulatory compliance to prevent severe harms like the leakage of financial or medical records.

*Machine unlearning* has emerged as a promising solution to address these privacy concerns by selectively removing specific information from trained models. While exact unlearning through retraining is computationally infeasible for LLMs, researchers developed approximate unlearning methods, including training pipeline manipulation (e.g., GA (Jang et al., 2023), NPO (Zhang et al., 2024)) and data manipulation (e.g., Random labeling (Maini et al., 2024), WHP (Eldan & Russinovich, 2023)). These approaches have demonstrated effectiveness under current evaluation frameworks such as TOFU (Maini et al., 2024), MUSE (Shi et al., 2024b), and WMDP (Li et al., 2024).

However, existing unlearning benchmarks face a critical limitation: they primarily evaluate against passive attackers who only observe model outputs, failing to assess resilience against active attackers who can manipulate models through techniques like in-context learning (ICL) (Huang et al., 2022; Shumailov et al., 2024) and fine-tuning (Hu et al., 2024; Chen et al., 2024). This evaluation gap leaves fundamental questions unanswered about the true effectiveness of different unlearning approaches. While prior work shows that active attacks can recover private information, we lack systematic frameworks to: (1) *compare the robustness of different unlearning methods against active attacks*, (2) *understand the mechanisms behind incomplete unlearning*, and (3) *evaluate performance under realistic constraints where defenders have limited knowledge of private data.*

• *New evaluation benchmark.* To fill this gap, we propose PriLeak, the first comprehensive benchmark specifically designed to assess unlearning robustness against active attackers across three-tier attack scenarios: direct retrieval through question-answering (**P1**), recovery via ICL (**P2**), and recovery via fine-tuning (**P3**). Crucially, PriLeak evaluates on both known and unknown private data, addressing the realistic constraint where defenders access only a subset of private data.

Moreover, to explain why private information persists after unlearning, we develop three quantitative metrics: (i) forgetting scores that measure the divergence in generation probabilities for PII sequences, providing assessment of forgetting effectiveness; (ii) association metrics that capture relationships between data samples through gradients (optimization relationships) and representation features (hidden state relationships); (iii) forgetting depth assessment using layer-wise *Centered Kernel Alignment* (CKA) analysis to track representation changes across model layers and pinpoint where private information remains.

• *New understanding in privacy forgetting.* We conducted the first systematic comparison of unlearning robustness against active attacks across 19 approaches. Our evaluation reveals that training pipeline manipulation methods demonstrate superior resilience compared to data manipulation approaches, with untargeted methods significantly outperforming targeted ones. Representation-based methods like RMU show better resistance than label-based alternatives, though at utility costs.

Our quantitative analysis uncovered two findings explaining incomplete unlearning: (1) Unlearning exhibits a unique **ripple effects** across associated data. Crucially, PII entanglement is non-semantic, distinguishing it from the logical associations in general knowledge. Instead, it is driven by latent optimization dynamics, evidenced by a strong correlation (Pearson $r = 0.73$) between gradient-based association and forgetting effectiveness. (2) Most methods exhibit **shallow forgetting**, failing to remove private information distributed across multiple layers. Forgetting depth assessment using layer-wise CKA analysis reveals that these methods primarily modify final layers while leaving intermediate representations largely unchanged.

• *New strategy for improving unlearning.* Building on these insights, we propose two key strategies to enhance unlearning effectiveness: (1) Association-aware core-set selection based on gradient similarity achieves 32.19% **P3** recovery rate using only 10% as core forget set compared to random 50% selection. (2) Multi-layer deep intervention through progressive learning rates and representational constraints reduces **P3** to 35.03% while achieving significantly better utility than RMU.

These contributions represent a paradigm shift from shallow forgetting to deep forgetting, providing both a comprehensive evaluation framework for assessing unlearning robustness and key insights for developing more effective privacy protection methods in LLMs.

## 2 PRELIMINARIES

### 2.1 PRIVACY UNLEARNING FOR LLMs

LLMs trained on web-scale data inevitably memorize personally identifiable information (PII) such as home address and emails, because thoroughly filtering these massive datasets is a significant challenge. This memorization poses substantial privacy risks when these models are publicly deployed (Carlini et al., 2021; 2022; Huang et al., 2022; Lukas et al., 2023). To address this, machine unlearning aims to selectively remove private information from a trained model while preserving its overall performance on non-private data. This is typically achieved through fine-tuning methods that modify model parameters, trading theoretical guarantees of forgetting for computational efficiency.

Formally, let $f_{target}$ be a model trained on dataset $\mathbb{D} = (x_i, y_i)_{i \in [n]}$. The objective of unlearning is to remove the influence of a specific subset, the "forget set" $\mathbb{D}_F \subseteq \mathbb{D}$, while maintaining performance on the remaining "retain set" $\mathbb{D}_R = \mathbb{D} \backslash \mathbb{D}_F$. An unlearning algorithm $U$ creates a modified model $f_{unlearn} = U(f_{target}, \mathbb{D}_F, \mathbb{D}_R)$ that behaves as if trained from scratch only on $\mathbb{D}_R$. Crucially, this process must be significantly more efficient than full retraining while preserving model utility.

**Realistic Constraints.** Traditional machine unlearning settings motivated by GDPR (Politou et al., 2018) assume defenders have complete access to all data requiring removal. However, this proves unrealistic for LLMs trained on vast web-scale data, where private information is scattered throughout massive datasets and deeply intertwined with non-private content. Identifying and isolating every

sensitive data point becomes practically impossible. We therefore propose a more realistic formulation where defenders: (1) *have access to only a limited subset of private data requiring removal*; (2) *have white-box access to the target model $f_{target}$ trained on mixed private and non-private data.*

**Problem Definition.** Given $f_{target}$ trained on dataset $\mathbb{D}$ containing private data (forget set) $\mathbb{D}_F$ and non-private data (retain set) $\mathbb{D}_R$, and a known subset of private data $\mathbb{D}_{F_k} \subset \mathbb{D}$ (the "*known forget set*"), the defender's objective is to: (1) *remove the influence of entire private dataset $\mathbb{D}_F$, including both known ($\mathbb{D}_{F_k}$) and unknown ($\mathbb{D}_{F_{uk}} = \mathbb{D}_F \backslash \mathbb{D}_{F_k}$) components*; (2) *preserve model utility on non-private data $\mathbb{D}_R$*. The resulting model $f_{unlearn}$ should behave as if trained from scratch on $\mathbb{D}_R$.

## 2.2 LIMITATION OF UNLEARNING BENCHMARKS

The effectiveness of LLM unlearning is critically undermined by relearning attacks, where an adversary restores forgotten knowledge using just a few original samples (Hu et al., 2024). An attacker can achieve this through fine-tuning or even without model modification via in-context learning (ICL) (Shumailov et al., 2024).

This threat is severe in privacy scenarios, as sensitive information can be recovered with minimal data and publicly available tools (Chen et al., 2024). While defenses using techniques like sharpness-aware minimization (Fan et al., 2025) and adversarial training (Sheshadri et al., 2024) are emerging, they often fail to address these specific privacy risks (see Appendix B).

Existing unlearning benchmarks such as TOFU (Maini et al., 2024), MUSE (Shi et al., 2024b), and WMDP (Li et al., 2024) evaluate performance on diverse tasks, from synthetic Q&A to hazardous knowledge removal. However, these benchmarks share a critical limitation: they only test against passive attackers who observe model outputs. They fail to measure resilience against active attackers who strategically manipulate the model such as relearning attacks. This gap highlights the urgent need for a new benchmark testing unlearning robustness against such malicious interventions.

## 3 THE PriLeak EVALUATION BENCHMARK

### 3.1 THE DESIGN OF PriLeak

To address the limitation, we introduce PriLeak, a new benchmark that audits the robustness of unlearning methods against three active attacks and quantifies how completely private data is forgotten. **Evaluation Metrics.** PriLeak measure how successfully private data has been erased against three increasingly sophisticated active attack scenarios:

• **P1. Direct Retrieval** tests for direct memorization. We query the model with questions about the private data and measure the precision of its responses as $\frac{1}{|\mathbb{D}_F|}\Sigma_{q\in\mathbb{D}_F}\mathbf{1}_{\text{PII}\in f(q)}$, where $q$ is a question targeting PII from a sample in the forget set $\mathbb{D}_F$, and $\mathbf{1}_{\text{condition}}$ is 1 if the condition is true.

• **P2. Recovery via In-context Learning** assesses if private data can be recovered using few-shot prompts. The recovery rate is: $Rec_{ICL} = \frac{1}{|\mathbb{D}_F|}\Sigma_{q\in\mathbb{D}_F}\mathbf{1}_{\text{PII}\in f(q,k\text{-shot})}$, where $k$ is the number of examples needed for recovery, serving as a proxy for the attack cost.

• **P3. Recovery via Fine-tuning** evaluates the model's ultimate resistance to data recovery through fine-tuning. The recovery rate is: $Rec_{FT} = \frac{1}{|\mathbb{D}_F|}\Sigma_{q\in\mathbb{D}_F}\mathbf{1}_{\text{PII}\in f_{ft}(q)}$, where $f_{ft}$ represents the model after fine-tuning. The attack cost is measured by the data and computation required. The fine-tuning threat model aligns with real-world scenarios. For open-weight models, weight tampering (Che et al., 2025) and relearning (Hu et al., 2024) represent primary attack vectors. Even in black-box settings (e.g., GPT-4), attackers can exploit standard fine-tuning APIs to recover private data without direct parameter access.

We deliberately exclude the Min-K metric (Shi et al., 2024a) designed for *dataset inference* (i.e., was a dataset used for training), whereas our focus is on *PII extraction* (i.e., specific PII can be retrieved even via novel prompts). Our empirical results in Section C.2 confirms the limitation of Min-K.

To measure model utility, we define the **U1** metric, which is based on the C4 evaluation from MUSE (Shi et al., 2024b). Specifically, We assess the model's question-answering performance on the retain set $\mathbb{D}_R$ by computing the average ROUGE score between its generated answers and the ground-truth answers. A high U1 score indicates that the model's general capabilities have been well preserved.

**Evaluation Corpus.** For the forget set, we evaluate the privacy leakage on Enron (Klimt & Yang, 2004). This dataset contains approximately 500k emails from Enron Corporation employees, which were made public by the Federal Energy Regulatory Commission. For this corpus, we extract [PERSON, PII] pairs from the original texts to create a structured evaluation set, then generate question-answer pairs following a consistent template: "Q: Tell me the [*PII type*] of [PERSON], A: [PII]". This standardized format allows us to evaluate how well unlearning methods remove specific pieces of private information. Furthermore, to evaluate diffuse PIIs (implicit attributes in free-form text), we explicitly include the TOFU dataset (Maini et al., 2024) as an additional forget set. From the original corpus of 4,000 questions concerning fictitious authors, we extracted a targeted subset of 424 QA pairs focusing on Date of Birth and Birthplace of authors, which can be regarded as private information. Following Section 2.1, we divide each corpus into known and unknown forget sets. The known forget set comprises 5%-100% of the entire forget set. For clarity, we present results using 20% and 50% splits, though experiments with other proportions show similar trends.

For the retain set, we utilize the retain set from MUSE News (Shi et al., 2024b), which includes 3.56k news articles and can be processed into 100 question-answer pairs to evaluate whether the model retains its general capabilities after unlearning.

**Evaluated Unlearning Methods.** We survey a wide range of approximate unlearning methods, which can be broadly classified into two approaches. **Training pipeline manipulation** modifies the model's objective function to encourage forgetting, while **data manipulation** modifies the labels of the forget set to overwrite or confuse the model's knowledge. We augment these base methods with two common regularization techniques designed to preserve performance on the retain set: gradient-based descent (GDR) and KL divergence minimization (KLR).

This results in a comprehensive suite of 19 methods for evaluation. **Training pipeline manipulation** methods include: (1) Gradient Ascent (Jang et al., 2023) variants: GA, $\text{GA}_{\text{GDR}}$, $\text{GA}_{\text{KLR}}$; (2) NPO (Zhang et al., 2024) variants: NPO, $\text{NPO}_{\text{GDR}}$, $\text{NPO}_{\text{KLR}}$; (3) DPO (Zhang et al., 2024) variants: DPO, $\text{DPO}_{\text{GDR}}$, $\text{DPO}_{\text{KLR}}$; (4) Task Vector (Ilharco et al., 2023) and (5) RMU (Li et al., 2024). **Data manipulation** methods include: (1) Random Labeling/Mapping (Maini et al., 2024) variants[1]: RL, $\text{RL}_{\text{GDR}}$, RM, $\text{RM}_{\text{GDR}}$; (2) "I Don't Know" variants: IDK, $\text{IDK}_{\text{GDR}}$; (3) WHP (Eldan & Russinovich, 2023) variants: WHP, $\text{WHP}_{\text{GDR}}$.

Beyond the primary categorization, these methods can be further classified along two dimensions: (1) **Target vs Untargeted** methods (Yuan et al., 2025) differ in whether they specify alternative outputs (e.g., DPO, IDK) or simply aim to prevent original responses (e.g., NPO). (2) **Label-based vs Representation-based** methods differ in their intervention level: label-based methods modify output distributions, while representation-based methods like RMU directly alter hidden layer representations. The classification is shown in Table 1, and complete method details are provided in Section C.1.

As a gold-standard baseline, we include **Retrain Model** $f_{retrain}$, trained from scratch exclusively on the retain set. While computationally infeasible, this represents the ideal of any unlearning process.

## 3.2 QUANTIFYING FORGETABILITY

To understand why private information persists after unlearning, PriLeak evaluates the completeness of forgetting across three distinct metrics.

**Forgetting Score.** While **P3** recovery rate effectively reveals incomplete unlearning, it is unstable and sensitive to fine-tuning samples. Thus, we introduce a more robust *forgetting score*, which measures the divergence between the output distributions of the target and unlearned models. Notably, unlike prior work focused on general sequence prediction (Jang et al., 2023), our score specifically quantifies the change in generation probability for PII token sequences.

The sequential probability for a PII sequence $\mathbf{y} = [y_1, y_2, \ldots, y_T]$ given a prefix context $x$, is defined as the product of the conditional probabilities of generating each token in the sequence. Formally: $P_{seq}(y|x) = \prod_{t=1}^{T} P(y_t|x, y_1, \ldots, y_{t-1})$. For example, the email address "board@isda.org", might be tokenized as ["board", "@", "is", "da", ".org"], and its sequential probability would be the product of these five conditional probabilities. The forgetting score **FS** is then defined as the difference

---

[1]Since KLR operates on the loss function, it is only paired with training-pipeline manipulation methods.

in log-probabilities between the log-probabilities assigned to the PII sequence by the original target model $f_{target}$ and the unlearned model $f_{unlearn}$: $\text{FS}(x, y) = \log(P_{f_{target}}(y|x)) - \log(P_{f_{unlearn}}(y|x))$. A higher score signifies a larger drop in the sequence's generation probability after unlearning, indicating more effective forgetting.

**Association Score.** To understand how data points are computationally linked during unlearning, we introduce an *Association Score (AS)* that quantifies the relationships between samples through two perspectives: gradient-based and representation-based associations.

*Gradient-based Association* captures optimization relationships by measuring how similarly different samples influence the model's update directions. For any $x_i \in \mathbb{D}_{uk}$ and $x_j \in \mathbb{D}_k$, the score is the dot product between their gradients with respect to the model parameters $\theta$: $\text{AS}_{grad}(x_i, x_j) = \nabla_{\boldsymbol{\theta}} L(f(x_i; \boldsymbol{\theta})) \cdot \nabla_{\boldsymbol{\theta}} L(f(x_j; \boldsymbol{\theta}))$, where gradients are computed specifically on the loss of PII token sequences. This metric is closely related to Neural Tangent Kernel (NTK) similarity (Jacot et al., 2018), which characterizes data relationships in the optimization landscape. We use dot product rather than cosine similarity to preserve the magnitude of gradients, not just their directions. Higher scores indicate that samples push model parameters in similar directions with similar force, suggesting strong computational coupling during unlearning.

*Representation-based Association* measures semantic relationships through hidden state similarity. For $x_i \in \mathbb{D}_{uk}$ and $x_j \in \mathbb{D}_k$, we compute the cosine similarity between their representation features as: $\text{AS}_{repr}(x_i, x_j) = \cos(h_l(x_i), h_l(x_j))$, where $h_l(\cdot)$ represents the average hidden states at layer $l$ for PII tokens. For each sample in $\mathbb{D}_{uk}$, its association score with $\mathbb{D}_k$ is computed as the average of pairwise similarities across all samples in $\mathbb{D}_k$. This captures whether samples are forgotten due to semantic similarity in their internal representation space.

**Forgetting Depth Assessment.** To assess how deeply private data is removed from a model's layers, we propose a two-pronged approach that combines behavioral and representational analysis.

*Recovery Rate Gap (Behavioral Analysis)*: First, we quantify the depth of forgetting by the comparing recovery rates under two attack scenarios: direct retrieval (**P1**) and recovery via fine-tuning (**P3**). We conceptualize **P1** as a failure of *shallow forgetting* while **P3** indicates a failure of *deep forgetting*. A large gap between **P1** and **P3** suggests that while surface-level PII is removed, the underlying knowledge remains and can be restored with fine-tuning, indicating insufficient deep unlearning.

*CKA Layer-wise Analysis (Representational Analysis)*: Next, to pinpoint which specific layers are affected by unlearning, we use Centered Kernel Alignment (CKA) (Kornblith et al., 2019) to measure the similarity of learned representations between the target model $f_{target}$ and the unlearned model $f_{unlearn}$ on a layer-by-layer basis. A lower CKA score at a specific layer signifies a greater change in its representations, which suggests more effective forgetting at that depth. By plotting CKA scores across three specific layers (§4.3), we can visualize where the unlearning process had the most impact and where residual information might persist.

This dual-analysis framework allows us to both explicitly measure unlearning outcomes (the "what") and implicitly analyze the underlying layer-wise mechanisms (the "where"). It provides the systematic basis for the two key findings we present regarding the limitations of current methods.

# 4 UNDERSTANDING CHALLENGES OF FORGETTING

## 4.1 MEASUREMENT STUDY

**Experimental Setup.** Our experiments start with a pre-trained LLaMA-3.2-3B model (Dubey et al., 2024) as the base architecture. From this, we fine-tune two initial models for our analysis: the target model ($f_{target}$) trained for 5 epochs on the full dataset $\mathbb{D}_F \cup \mathbb{D}_R$, which combines samples from the Enron and MUSE News datasets, and the retrained model ($f_{retrain}$) trained using only the retain set $\mathbb{D}_R$, which serves as our gold standard. For each unlearning method $U$ being evaluated, we then generate an unlearned model $f_{unlearn}$ by applying $U$ to the target model: $f_{unlearn} = U(f_{target}, \mathbb{D}_{F_k}, \mathbb{D}_R)$. All unlearning methods use a learning rate of $10^{-5}$ and batch size of 32, with hyperparameters chosen to maximize utility on a validation set. To ensure reliability, we average all results over 3 runs with different random seeds, conducted on 4 NVIDIA A100 GPUs.

Table 1: Privacy and utility measurements on Enron and News using LLaMA-3.2. For $f_{target}$ and $f_{retrain}$, recovery rates are evaluated on the full $\mathbb{D}_F$ as they do not have forget set splits. **P3**'s recovery cost uses $|D|$ for fine-tuning data size and $e$ for epochs. **U1** uses 10-shot in-context learning. For the classification, training pipeline or data manipulation; targeted (✓) or untargeted (✗); label-based (✓) or representations-based (✗).

| Methods | Classification | | P1. Direct Retrieval | | P2. Recovery via ICL | | P3. Recovery via Fine-tuning | | | U1. Utility Preserv. | |
| | Targeted | Label-based | $Rec$ on $\mathbb{D}_{F_k}$ | $Rec$ on $\mathbb{D}_{F_{uk}}$ | $Rec$ on $\mathbb{D}_{F_k}$ | $Rec$ on $\mathbb{D}_{F_{uk}}$ | $Rec$ on $\mathbb{D}_{F_k}$ | $Rec$ on $\mathbb{D}_{F_{uk}}$ | Cost | QA | QA+ICL |
|---|---|---|---|---|---|---|---|---|---|---|---|
| Target $f_{target}$ | | | 14.53% | | 29.94% | | 60.60% | | $|D|=50, 3e$ | 27.35 | 36.38 |
| Retrain $f_{retrain}$ | | | 0.02% | | 4.95% | | 13.14% | | $|D|=50, 3e$ | 26.69 | 35.04 |
| **20% Enron + News** | | | | | | | | | | | |
| GA | ✗ | ✓ | 0.49% | 0.43% | 0.49% | 0.61% | 54.27% | 54.68% | $|D|=50, 10e$ | 26.71 | 36.19 |
| GA$_{GDR}$ | ✗ | ✓ | 0.24% | 0.43% | 0.37% | 0.47% | 53.10% | 53.30% | $|D|=50, 10e$ | 27.57 | 34.27 |
| GA$_{KLR}$ | ✗ | ✓ | 12.20% | 13.13% | 31.34% | 32.84% | 66.10% | 64.58% | $|D|=50, 3e$ | 27.35 | 48.25 |
| NPO | ✗ | ✓ | 0.49% | 0.49% | 0.24% | 0.06% | 56.83% | 58.22% | $|D|=50, 3e$ | 23.78 | 36.48 |
| NPO$_{GDR}$ | ✗ | ✓ | 0.37% | 0.52% | 0.24% | 0.06% | 57.56% | 57.42% | $|D|=50, 3e$ | 24.55 | 37.49 |
| NPO$_{KLR}$ | ✗ | ✓ | 12.20% | 13.13% | 31.34% | 32.84% | 64.58% | 64.58% | $|D|=50, 3e$ | 27.35 | 36.38 |
| Task Vector | ✗ | ✓ | 2.92% | 2.62% | 12.68% | 12.82% | 66.10% | 64.58% | $|D|=50, 3e$ | 26.49 | 37.05 |
| DPO | ✓ | ✓ | 0.00% | 0.00% | 30.61% | 31.16% | 65.98% | 65.46% | $|D|=50, 10e$ | 9.97 | 24.76 |
| DPO$_{GDR}$ | ✓ | ✓ | 0.00% | 0.00% | 34.27% | 35.09% | 69.51% | 67.86% | $|D|=50, 10e$ | 10.02 | 24.83 |
| DPO$_{KLR}$ | ✓ | ✓ | 0.00% | 0.06% | 51.10% | 51.11% | 65.24% | 63.81% | $|D|=50, 3e$ | 11.69 | 22.84 |
| RMU | ✗ | ✗ | 0.00% | 0.00% | 0.00% | 0.00% | 18.32% | 20.30% | $|D|=50, 10e$ | 8.35 | 9.51 |
| RL | ✗ | ✓ | 12.32% | 13.00% | 37.32% | 37.95% | 65.85% | 64.64% | $|D|=50, 3e$ | 29.25 | 36.36 |
| RL$_{GDR}$ | ✗ | ✓ | 12.44% | 12.03% | 39.63% | 37.56% | 62.93% | 60.83% | $|D|=50, 3e$ | 31.33 | 34.07 |
| RM | ✗ | ✓ | 30.73% | 28.33% | 38.05% | 37.22% | 25.85% | 21.81% | $|D|=50, 3e$ | 29.09 | 34.67 |
| RM$_{GDR}$ | ✗ | ✓ | 40.73% | 38.87% | 36.10% | 37.07% | 65.12% | 63.08% | $|D|=50, 3e$ | 32.31 | 35.55 |
| WHP | ✗ | ✓ | 1.59% | 0.70% | 22.68% | 23.09% | 66.71% | 65.28% | $|D|=50, 3e$ | 28.76 | 36.09 |
| WHP$_{GDR}$ | ✗ | ✓ | 0.98% | 0.82% | 12.32% | 12.98% | 65.85% | 66.49% | $|D|=50, 10e$ | 27.49 | 35.38 |
| IDK | ✓ | ✓ | 16.71% | 15.35% | 32.56% | 34.05% | 67.68% | 66.43% | $|D|=50, 3e$ | 28.95 | 35.95 |
| IDK$_{GDR}$ | ✓ | ✓ | 21.83% | 20.07% | 29.63% | 29.61% | 66.71% | 66.31% | $|D|=50, 3e$ | 33.49 | 34.99 |
| **50% Enron + News** | | | | | | | | | | | |
| GA | ✗ | ✓ | 0.39% | 0.49% | 0.44% | 0.39% | 30.82% | 32.47% | $|D|=50, 10e$ | 26.00 | 35.02 |
| GA$_{GDR}$ | ✗ | ✓ | 0.34% | 0.39% | 0.44% | 0.34% | 39.83% | 41.20% | $|D|=50, 10e$ | 28.99 | 34.97 |
| GA$_{KLR}$ | ✗ | ✓ | 12.09% | 13.80% | 32.62% | 32.47% | 64.31% | 65.48% | $|D|=50, 3e$ | 27.35 | 36.38 |
| NPO | ✗ | ✓ | 0.34% | 0.44% | 0.10% | 0.05% | 56.61% | 56.61% | $|D|=50, 3e$ | 20.07 | 38.15 |
| NPO$_{GDR}$ | ✗ | ✓ | 0.44% | 0.54% | 0.10% | 0.05% | 56.12% | 58.75% | $|D|=50, 3e$ | 25.33 | 36.91 |
| NPO$_{KLR}$ | ✗ | ✓ | 12.09% | 13.80% | 32.62% | 32.47% | 38.27% | 38.86% | $|D|=50, 3e$ | 27.35 | 36.38 |
| Task Vector | ✗ | ✓ | 3.94% | 4.29% | 18.04% | 17.60% | 64.75% | 65.77% | $|D|=50, 3e$ | 27.48 | 37.41 |
| DPO | ✓ | ✓ | 0.00% | 0.00% | 0.00% | 0.00% | 67.92% | 69.43% | $|D|=50, 10e$ | 0.00 | 6.44 |
| DPO$_{GDR}$ | ✓ | ✓ | 0.00% | 0.00% | 1.37% | 1.66% | 63.04% | 64.16% | $|D|=50, 10e$ | 0.00 | 15.19 |
| DPO$_{KLR}$ | ✓ | ✓ | 0.00% | 0.06% | 58.65% | 60.99% | 65.53% | 67.09% | $|D|=50, 3e$ | 11.81 | 19.94 |
| RMU | ✗ | ✗ | 0.00% | 0.00% | 0.00% | 0.00% | 16.52% | 18.14% | $|D|=50, 10e$ | 7.75 | 8.68 |
| RL | ✗ | ✓ | 9.56% | 11.41% | 43.49% | 43.98% | 62.90% | 63.43% | $|D|=50, 3e$ | 30.12 | 35.44 |
| RL$_{GDR}$ | ✗ | ✓ | 10.04% | 9.90% | 31.25% | 29.50% | 52.80% | 53.24% | $|D|=50, 3e$ | 30.23 | 34.88 |
| RM | ✗ | ✓ | 27.89% | 29.84% | 39.05% | 40.03% | 62.46% | 62.51% | $|D|=50, 3e$ | 28.78 | 35.99 |
| RM$_{GDR}$ | ✗ | ✓ | 33.40% | 34.42% | 20.09% | 19.80% | 62.46% | 62.46% | $|D|=50, 3e$ | 30.77 | 35.91 |
| WHP | ✗ | ✓ | 0.59% | 0.68% | 18.97% | 19.99% | 61.87% | 64.65% | $|D|=50, 3e$ | 27.50 | 35.68 |
| WHP$_{GDR}$ | ✗ | ✓ | 0.15% | 0.24% | 4.29% | 5.27% | 61.87% | 62.85% | $|D|=50, 10e$ | 24.57 | 33.73 |
| IDK | ✓ | ✓ | 13.65% | 15.02% | 32.91% | 32.96% | 65.68% | 67.28% | $|D|=50, 3e$ | 28.95 | 35.95 |
| IDK$_{GDR}$ | ✓ | ✓ | 19.25% | 21.11% | 30.91% | 31.69% | 65.82% | 50.76% | $|D|=50, 3e$ | 29.97 | 34.81 |

We verified that our findings are consistent on a GPT-2-Large model (Radford et al., 2019), with those results available in Section D.2. The implementation details are described in Section D.1.

**Effectiveness against Passive Observation (P1).** *Training Pipeline vs Data Manipulation Methods* demonstrate different effectiveness patterns when evaluated on direct retrieval **P1**. Training pipeline manipulation methods (GA, NPO) are highly effective, achieving near-perfect protection with information leakage rates below 0.5%. In contrast, data manipulation methods exhibit highly inconsistent behavior. Random labeling variants (RL, RM, IDK) lead to more PII leakage than the target model, with the worst performer (RM) reaching a 40.73% recovery rate. This suggests that simply providing alternative labels may fail to erase the underlying memorized patterns and can sometimes even reinforce them. This performance gap stems from a fundamental difference in mechanisms. Training-pipeline manipulation methods directly alter the model's optimization objective to make generating private information less likely. Data manipulation methods, however, merely attempt to overwrite existing knowledge with new targets. Our results show the former is a far more reliable strategy for basic privacy protection, though its robustness against more advanced active attacks remains to be examined.

The choice of regularization strategy also significantly impacts unlearning effectiveness. Methods using KL regularization (GA$_{KLR}$, NPO$_{KLR}$, DPO$_{KLR}$) show substantially higher leakage rates compared to their counterparts using GDR. This performance gap suggests a critical trade-off. While KLR is designed to preserve the model's overall output distribution, this conservative approach may inadvertently protect the very knowledge pathways that allow for PII retrieval. In contrast, GDR's superior performance appears to stem from its more targeted constraint on the optimization process, which drives the model to more aggressively unlearn the specific PII.

**Resilience to Active Attacks (P2&P3).** Most unlearning methods are vulnerable to active attacks, particularly pronounced under **P3**. A clear divide in robustness emerges between *targeted* and *untargeted* unlearning methods. Untargeted methods show far superior resilience to active attacks. For

Table 2: **Generalization to Diffuse PIIs (TOFU).** Recovery rates of representative methods evaluated on a 20% known forget set. Consistent with findings on Enron, methods that appear effective under **P1** remain highly vulnerable to **P3**.

| Methods | P1. Direct Retrieval | | P2. Recovery via ICL | | P3. Recovery via Fine-tuning | | |
|---|---|---|---|---|---|---|---|
| | $Rec$ on $\mathbb{D}_{F_k}$ | $Rec$ on $\mathbb{D}_{F_{uk}}$ | $Rec$ on $\mathbb{D}_{F_k}$ | $Rec$ on $\mathbb{D}_{F_{uk}}$ | $Rec$ on $\mathbb{D}_{F_k}$ | $Rec$ on $\mathbb{D}_{F_{uk}}$ | Cost |
| Target $f_{target}$ | 42.45% | | 47.64% | | 41.75% | | $|D| = 50, 1e$ |
| GA | 0.00% | 0.00% | 0.00% | 0.00% | 26.19% | 8.82% | $|D| = 50, 10e$ |
| NPO | 5.95% | 5.88% | 17.86% | 10.00% | 38.10% | 17.35% | $|D| = 50, 10e$ |
| DPO | 0.00% | 0.00% | 0.00% | 0.00% | 32.14% | 12.35% | $|D| = 50, 10e$ |
| RM | 10.71% | 0.29% | 9.52% | 1.47% | 35.71% | 3.53% | $|D| = 50, 10e$ |
| IDK | 0.00% | 0.00% | 0.00% | 0.00% | 27.38% | 7.06% | $|D| = 50, 10e$ |

instance, NPO keeps **P2** recovery rate near zero (0.2%), while the targeted method DPO is easily compromised, with a **P2** rate of over 30%. This suggests that forcing a model toward a specific alternative answer (e.g., "sorry I don't know") creates brittle changes that are easily reversed.

A similar distinction appears between *representation-based* and *label-based* methods. RMU consistently outperforms label-based alternatives, achieving **P3** of 18-20% compared to 60%+ for most others. This advantage stems from its mechanism: RMU directly modifies the model's hidden-layer representations to erase information, creating a deeper forgetting that is more durable than simply changing the output distribution, a phenomenon we explore further in Section 4.3.

To verify that PriLeak serves as a standard independent of hyperparameter tuning, we performed sensitivity analysis on attacker configurations. For P2, varying the shot count $k \in \{5, 10, 20\}$ yielded consistent recovery rates (e.g., GA fluctuated marginally between 0.49% and 0.53%). For P3, we varied the attacker's dataset size (10, 20, 50 samples). Results stabilized at 50 samples, indicating that attack success is driven by data availability rather than stochastic tuning.

**Parallel Forgetting Patterns.** A notable similarity in unlearning effectiveness emerges between the known forget set $\mathbb{D}_{F_k}$ and the unknown forget set $\mathbb{D}_{F_{uk}}$. Under $\mathsf{GA}_{\mathsf{GDR}}$ method, for instance, the **P3** recovery rate was nearly identical for both sets known data (53.10% for the known set and 53.30% for the unknown set) data, even though only the known set is subjected for removal. This effect becomes even more counter-intuitive with random labeling methods on GPT-2 (detailed in Appendix Table 3), where unknown forget data shows better forgetting effectiveness than the known set, creating performance gaps up to 7%. This parallel behavior suggests that unlearning specific data causes a systematic propagation of forgetting that extends beyond the explicit targets. We term this phenomenon the "ripple effect" and analyze it in detail in Section 4.2.

**Generalization to Diffuse PIIs.** To verify whether our findings extend beyond structured records (e.g., email-name pairs) to diffuse PIIs where private attributes like birthplaces are implicitly embedded in free-form narratives, we conducted additional evaluations on the TOFU benchmark. As shown in Table 2, the results align consistently with our findings on Enron: current unlearning methods fail to fully erase implicit private information. For example, while GA effectively suppresses direct retrieval (P1 reduced to 0%), it remains highly vulnerable to fine-tuning attacks, with the P3 recovery rate rebounding to 26.19%. NPO exhibits even severe leakage, maintaining a 38.10% recovery rate under P3. This confirms that the vulnerability is not an artifact of structured data but a fundamental limitation of current unlearning mechanisms against active attacks.

**Utility Preservation (U1).** LLaMA-3.2 is highly robust to utility degradation, with most unlearning methods maintaining QA performance close to the baseline QA performance (27.35). The notable exception is DPO and RMU, causing a more significant drop in performance. In contrast, GPT-2-Large suffers severe utility loss (Section D.2). This suggests that newer model architectures may be inherently better at preserving core capabilities during the unlearning process.

These results raised two critical questions: (1) *Why does unlearning a known set of data also cause forgetting in an unknown (but related) set?* (2) *Why do unlearning methods that appear effective under simple observation fail against active attacks?* In the following sections, we provide mechanistic explanations for these phenomena through two key findings: ripple effect and shallow forgetting.

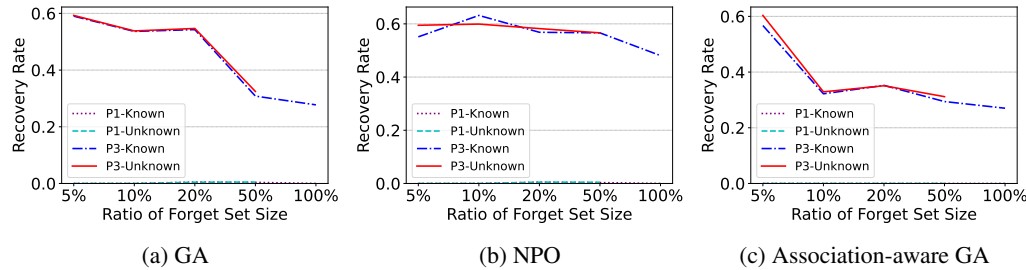

Figure 1: Recovery rate comparison between known and unknown forget sets across different forget set sizes and attack scenarios (P1, P3); (c) shows the effectiveness using association-aware core-set.

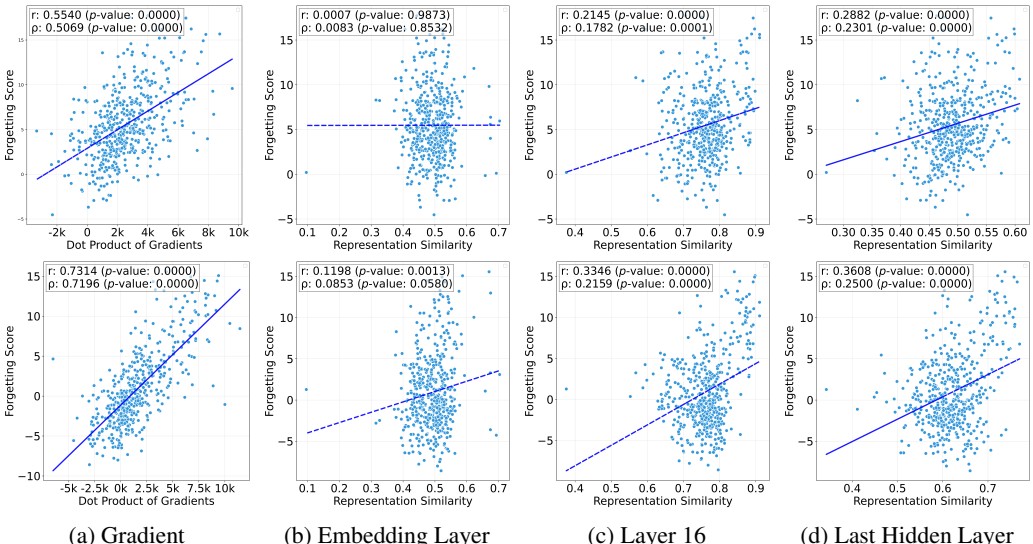

Figure 2: Correlation between gradient-based association (a), multi-layer representation-based association (b)-(d) and forgetting scores for NPO (top row) and GA (bottom row) methods ($r$ = Pearson coefficient; $\rho$ = Spearman coefficient).

## 4.2 RIPPLE EFFECT IN UNLEARNING

Our first key finding reveals that unlearning exhibits ripple effects across related data through shared neural representations. This phenomenon makes it fundamentally impossible to selectively forget specific data while preserving related information.

**Forget Set Size.** We first conduct experiments using varying proportions of the forget set: 5%, 10%, 20%, 50%, and 100%. This allows us to observe how forgetting effectiveness changes as more related data is included in the forget set. Figure 1 demonstrates the parallel forgetting patterns between $\mathbb{D}_k$ and $\mathbb{D}_{uk}$ across GA and NPO. As we increase the known forget set size, both sets exhibit remarkably similar recovery rate trajectories, providing direct evidence for the ripple effect.

Moreover, the ripple effect performs differently at various attack scenarios. For **P1**, even with minimal forget sets (5% Enron), methods achieve near-perfect recovery rates (below 1%) for both sets, indicating effective shallow ripple effects. However, for **P3**, both sets exhibit high recovery rates (50-70%) because these methods fail to remove residual information from deeper representations. This disparity shows that while shallow forgetting benefits from ripple effect, deep forgetting faces limitations that affect both forget set and associated data equally, as further explored in Section 4.3.

**Gradient-based Association.** To understand the ripple effect, we analyze the relationship between gradient similarity and forgetting effectiveness. For each sample in $\mathbb{D}_{F_{uk}}$, we compute its gradient's similarity to the average gradient of $\mathbb{D}_{F_k}$ and correlate this with its forgetting score.

Figure 2a presents a scatter plot showing the relationship between gradient association scores (x-axis) and forgetting effectiveness (y-axis) for $\mathbb{D}_{uk}$. For GA, the plot (bottom row) reveals a strong positive correlation (Pearson coefficient: 0.7314, Spearman: 0.7196), demonstrating that samples

with higher gradient similarity consistently achieve higher forgetting scores. This pattern provides concrete evidence that forgetting propagation is driven by gradient-based associations rather than random effects. Samples sharing similar optimization directions with the $\mathbb{D}_k$ are more likely to be forgotten. NPO exhibits a similar correlation pattern (Pearson: 0.5540, Spearman: 0.5069), confirming that this gradient-driven ripple effect is consistent across different unlearning methods.

**Representation-based Association.** To further validate the findings, we examine layer-wise representation similarity between samples in $\mathbb{D}_{F_k}$ and $\mathbb{D}_{F_{uk}}$. For each sample in $\mathbb{D}_{F_{uk}}$, we analyze the correlation between its representation similarity and forgetting score to understand how semantic embeddings relate to forgetting propagation.

Figure 2(b)-(d) reveals three key findings about the relationship between representational similarity and the ripple effect: (1) **No Link at Surface Layers:** At the initial embedding layer, we find no meaningful correlation, indicating that the ripple effect is not driven by surface-level semantic similarity. (2) **Association Emerges in Deeper Layers:** The correlation begins to emerge around layer 16, suggesting that the data associations responsible for co-forgetting are formed during deeper semantic processing. (3) **Optimization Dynamics Dominate:** The maximum correlation based on representations is weak (Pearson's $r = 0.3608$ at the last hidden layer), significantly lower than the correlation based on gradients ($r = 0.7314$), indicating that gradient-based optimization dynamics, not simply representational similarity, are the primary driver of the ripple effect.

These findings mark a fundamental distinction from concurrent works on knowledge unlearning (Wu et al., 2024; Wei et al., 2025) and knowledge editing (Cohen et al., 2024). While those studies characterize association based on semantic knowledge graphs (e.g., logical entailment), our results demonstrate that PII entanglement is driven by latent optimization dynamics. Unlike general knowledge, PIIs are linked via shared parameter update directions rather than human-interpretable semantic webs.

**Strategy: Association-aware Core-set Selection.** Building on this finding, we propose a strategic shift from random sampling to association-aware core-set selection. Rather than treating all samples in the forget set equally, we leverage the ripple effect by prioritizing samples with the highest associative influence, involving three steps: (1) Compute the mean gradient vector across all samples in the forget set $\mathbb{D}_F$ to establish the representative gradient pattern: $\bar{g} = \frac{1}{|\mathcal{D}_F|} \sum_{x \in \mathcal{D}_F} \nabla_\theta L(f(x; \boldsymbol{\theta}))$. (2) Compute the association score of each sample $x \in \mathcal{D}_F$: $\text{AS}(x) = \nabla_\theta L(f(x; \boldsymbol{\theta})) \cdot \bar{g}$. (3) Select the top k% samples with highest gradient association scores as the core forget set.

We evaluated core-sets of 5%, 10%, 20%, and 50% using GA unlearning. Figure 1c shows that its ripple effect is more effective under **P3** compared to random set: a 10% core-set achieves a **P3** recovery rate of 32.19%, matching the performance of random 50% and full forget sets. Therefore, instead of attempting selective removal while preserving associated data, the core-set method strategically exploits the ripple effect to achieve comprehensive privacy removal with minimal data.

## 4.3 SHALLOW FORGETTING

Another observation reveals that existing unlearning methods suffer from "shallow forgetting": they fail to resist active attackers because private information persists across hidden layers. We analyze this phenomenon by comparing different unlearning methods along two dimensions: attack-resistance gaps (**P1** vs **P3**) and layer-wise representation changes.

**Attack Resistance Gap Comparison.** Figure 3 presents the P1 and P3 recovery rates across different unlearning methods, revealing three distinct patterns in unlearning effectiveness: (1) methods achieving both shallow and deep forgetting with modest gaps (RMU), (2) methods failing across all scenarios (RL and IDK variants), and (3) methods demonstrating severe shallow forgetting with dramatic P1-P3 disparities up to 69.5% (DPO, NPO, GA variants, Task Vector), highlighting the persistence of private information in deeper model representations despite shallow protection. The analysis reveals that 15 out of 19 methods exhibit gaps exceeding 50%, indicating that shallow forgetting is a fundamental limitation of current unlearning rather than a method-specific issue.

**Layer-wise Representation Comparison.** To understand where privacy information persists after unlearning, we employ CKA analysis to compare how different unlearning methods modify representations across model layers. Figure 4 presents the CKA similarity across all 28 layers (lay 0 represents embedding layer) for both training pipeline manipulation methods (left) and data manip-

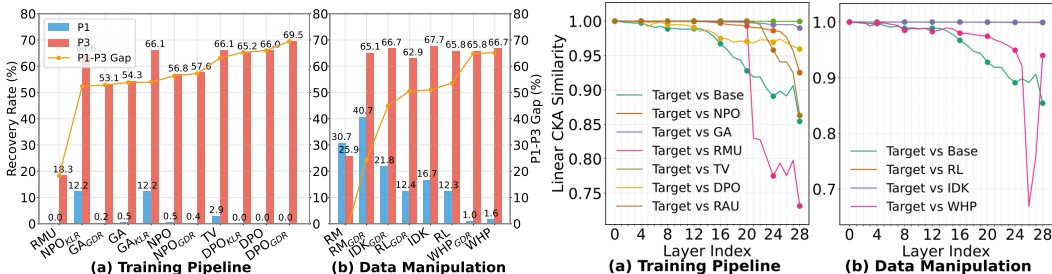

Figure 3: Shallow forgetting analysis across unlearning methods using 20% Enron dataset.

Figure 4: Cross-model CKA analysis among representative unlearning methods.

ulation methods (right). While data manipulation methods show minimal deviation from the target model across most layers, training pipeline methods exhibit more significant representation changes. This gap between the two categories aligns with our benchmarking results presented in Section 4.1. Moreover, label-based methods only show representation changes near the output layers (the best-performing method, NPO, maintains similarity over 0.98 until a sharp decline at layer 24). In contrast, the representation-based method RMU shows the most aggressive changes (dropping to 0.83 at layer 20), which, however, results in utility collapse (8.35 in Table 1).

This layer-wise analysis reveals that overly aggressive changes harm utility while output-layer modifications cause shallow forgetting. Therefore, unlearning methods should focus on inducing *moderate changes in mid-layer representation* that balance deep forgetting and model utility preservation.

**Strategy: Multi-layer Intervention.** Based on our findings, effective unlearning requires targeting hidden layer representations with increased learning efforts for deeper layers and loss functions that address privacy removal at multiple network depths. RMU outperforms other methods because it directly targets hidden representations rather than just output distributions. However, Figure 4 and Table 1 show that excessive changes toward random control vectors destroy model utility. To address this limitation, we propose **Representation Anchoring Unlearning** (RAU), which builds on RMU with two key improvements:

(1) Implement depth-dependent learning rates: $\eta_l = \eta \cdot \gamma^{l_{out} - l}$ where $\eta_l$ is the learning rate for layer $l$, $l_{out}$ is the output layer, and $\gamma > 1$ is a scaling factor, ensuring stronger updates in deeper layers.

(2) Instead of random control vectors, we anchor representations to noise-perturbed base model states: $\mathcal{L}_{anchor} = \sum_{l=l_0}^{l_{out}} \alpha_l ||h_l^{target} - (h_l^{base} + \epsilon)||_2^2$ where $h_l$ represents hidden states at layer $l$, $\epsilon \sim \mathcal{N}(0, \sigma^2)$ is Gaussian noise, and $\alpha_l$ are layer-specific weights. The final loss combines unlearning and utility preservation, i.e., $\lambda_{unlearn}\mathcal{L}_{anchor} + \lambda_{retain}\mathcal{L}_{retain}$.

Using layers 20-28, RAU achieves **P3** recovery rates of 35.03% on $\mathbb{D}_k$ and 38.55% on $\mathbb{D}_{uk}$ with 20% forget sets. While higher than RMU's recovery rates, RAU maintains significantly better model utility (27.70 and 35.41), achieving optimal trade-off between privacy protection and utility preservation. Figure 4 shows that RAU begins moderate representation changes at Layer 20 and converges identically with the base model at Layer 28. This approach represents a paradigm shift from *shallow forgetting* to *deep forgetting*, addressing private information distributed across multiple layers.

## 5 CONCLUSION

This work fills a critical gap in evaluating LLM unlearning under active privacy attacks. PriLeak shows that while current methods seem effective against passive observation, they remain vulnerable to active attackers who can recover forgotten private information. Our evaluation of 19 methods reveals two findings: ripple effect, where unlearning propagates across related data, and shallow forgetting, where private information persists in deeper layers. PriLeak's quantitative analysis offers the first mechanistic understanding of privacy persistence, highlighting the need for association-aware, multi-layer strategies beyond output-focused unlearning to achieve robust privacy protection.

ETHICS STATEMENT

In our research, we conducted all experiments using public datasets and models. However, it is still possible to extract real-world private information, such as phone numbers and home addresses, from the training data. To mitigate ethical risks, any extracted PII was promptly deleted after being compared with ground truth data. Additionally, we sought approval from our institution's IRB, which confirmed that "no human subjects are involved" and approved our study.

REPRODUCIBILITY STATEMENT

All data and programs used in the evaluation section are publicly available. Furthermore, the authors will fully disclose the source code, manipulated datasets, and other benchmark tools as a standalone artifacts to the public to support future research. Artifacts can be reviewed at: `https://anonymous.4open.science/r/PII_Unlearning-0D08/`.

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

## A   USE OF LLMS

This paper utilized Claude (Anthropic) and GPT-4 (OpenAI) as writing assistance tools to enhance clarity and readability of the manuscript. The LLMs were employed specifically for refining sentence structure and polishing writing throughout the paper. All research ideas, experimental design, data analysis, and scientific conclusions remain entirely the work of the authors. The LLMs did not contribute to methodology development or interpretation of results. Authors take full responsibility for the accuracy and integrity of all content, including any text refined with LLM assistance.

## B   RELATED WORK

**Robust Unlearning.** A line of works identified that current LLM unlearning methods face significant robustness challenges, with unlearned knowledge being recoverable through various attacks (Łucki et al., 2025). Che et al. (2025) evaluated LLM with model tampering attacks, which allows modifications to latent activations or weights, Zhang et al. (2025) revealed catastrophic failures of unlearning via quantization, and Lynch et al. (2024) emphasized that current approaches lack robustness against adversarial threats. Hu et al. (2024) introduced the relearning attack, which provides an efficient means of recovering forgotten data. In response, some works hae sought to strengthen the robustness of unlearning methods, For instance, Fan et al. (2025) leveraged Sharpness-Aware Minimization (SAM) to improve unlearning performance, while Zhang et al. (2025) proposed a tailored framework to counteract the effects of model quantization in the unlearning task.

While these studies share our observation that forgotten knowledge often persists in intermediate layers, our work differs in focus and methodology. Prior efforts primarily highlight the limited generalization of unlearning techniques, demonstrating that forgotten knowledge can be recovered through some adversarial attacks. In contrast, we assess the effectiveness of unlearning under three

progressively challenging attack settings, each probing different degrees of information retention. Based on these insights, we introduce a framework that builds upon existing approaches to achieve more comprehensive privacy removal. This offers a new paradigm for enhancing the robustness of unlearning in large models.

## C  PriLeak COMPONENTS

### C.1  EVALUATED UNLEARNING METHODS

While exact unlearning through retraining is theoretically optimal, it is computationally infeasible for LLMs. Therefore, our research focuses on approximate unlearning methods, which can be broadly classified into two main categories: training pipeline manipulation and data manipulation.

**Training Pipeline Manipulation.** Training pipeline manipulation methods modify the training loss or model parameters to remove unwanted information. Gradient ascent (GA) (Jang et al., 2023) minimizes the likelihood of correct predictions on forget set by performing gradient ascent on cross-entropy loss ($\ell_{GA}$):

$$\mathcal{L}_{GA}(\theta) = -\mathbb{E}_{(x,y) \in D_F}[-log(f_\theta(y|x))] \tag{1}$$

where $\theta$ represents the model parameters to be updated during unlearning. The rationale of GA is that a maximization of prediction loss on the forget set $\mathbb{D}_F$ would approximately "revert" the optimization on $\mathbb{D}_F$.

Negative Preference Optimization (NPO) (Zhang et al., 2024) treats the forget set as negative preference data, and adapts the objective for offline Direct Preference Optimization (DPO). Unlike GA's unbounded loss, NPO transforms the objective into a bounded loss:

$$\mathcal{L}_{NPO}(\theta) = -\frac{2}{\beta}\mathbb{E}_{(x,y) \in D_F}[log\sigma(-\beta log\frac{f_\theta(x)}{f_{target}(x)})] \tag{2}$$

where $\sigma$ is the sigmoid function, and $\beta$ controls divergence from its original model $f_{target}$. This formulation provides more controlled and stable unlearning compared to the straightforward gradient ascent method.

Representation Misdirection for Unlearning (RMU) (Li et al., 2024) is a fine-tuning based unlearning method inspired by representation engineering that operates by steering the model's internal representations. The RMU objective optimizes the following MSE loss:

$$\mathcal{L} = \mathbb{E}x_F \in \mathcal{D}_F||h_{\theta^{unlearn}}^{(l)}(x_F) - cu||_2^2 \tag{3}$$

$$+ \alpha\mathbb{E}x_R \in \mathcal{D}_R||h\theta^{unlearn^{(l)}}(x_R) - h_{\theta^{frozen}}^{(l)}(x_R)||_2^2 \tag{4}$$

where $\theta^{unlearn}$ and $\theta^{frozen}$ are parameters of the updated model and frozen model respectively, $u$ is a fixed random unit vector, $c$ is a scaling factor, $l$ denotes the target layer, and $\alpha$ balances the two objectives.

Besides modifying loss function, certain model editing techniques directly adjust the model parameters. One such example is the Task Vector method (Ilharco et al., 2023), which alters the training trajectory by editing model weights with task arithmetic:

$$f_{unlearn} = f_{target} - \lambda(f_{reinforce} - f_{target}) \tag{5}$$

where $f_{reinforce}$ is obtained by overfitting on forget set, and $\lambda$ is a scaling term.

**Data Manipulation.** Data manipulation modifies the training data or their labels to achieve unlearning. The simplest approach is Random Labeling (RL) (Maini et al., 2024), which relabels the samples in forget set with random (but seemingly sensible) outputs to force unlearning of original associations. Similarly, Random Mapping (RM) randomly pairs the existing inputs and labels, and "I don't know" (IDK) strategy replaces target outputs with uncertainty indicators. The new label is denoted as $y'$:

$$\mathcal{L}_{RL}(\theta) = \mathbb{E}_{(x,y) \in D_F}[logf_\theta(y'|x)] \tag{6}$$

Who's Harry Potter (WHP) (Eldan & Russinovich, 2023) provides a more sophisticated data manipulation strategy. It generates the output distribution of the unlearned model $f_{unlearn}$ by interpolating

between a reinforced model's predictions and the target model's predictions:

$$p_{f_{unlearn}}(\cdot|x) = p_{f_{target}}(\cdot|x) - \alpha(p_{f_{reinforce}}(\cdot|x) - p_{f_{target}}(\cdot|x)) \qquad (7)$$

where $p_f(\cdot|x)$ denotes the token distribution when given a prompt $x$ as input, and $\alpha$ controls the interpolation strength. Then it samples the alternative labels $y'$ from this interpolated distribution.

$$\mathcal{L}_{WHP}(\theta) = \mathbb{E}_{(x,y)\in D_F}[log f_\theta(y'|x)], y' \sim p_{f_{unlearn}}(\cdot|x) \qquad (8)$$

This approach creates alternative training targets that help remove specific information while preserving general language capabilities.

## C.2 Evaluation Metrics: Why Not Min-k

We didn't adopt Min-K (Shi et al., 2024a) here as a metric because it cannot serve as a gold standard for privacy unlearning evaluation. Min-K is used to test whether a specific training sample was used for model training, while we only focus on whether specific PII can be extracted from the model (e.g., through various queries of token sequences not present in the training set). Our empirical evaluation confirms this limitation of Min-K. For the forget set, the target model achieves an AUC of 0.436 and the unlearned model shows 0.424, both below the random baseline of 0.5. This indicates that Min-K performs worse than random chance in distinguishing between member and non-member samples in both models. More critically, the negligible difference between target and unlearned models (0.436 vs 0.424) demonstrates that Min-K cannot detect whether unlearning has occurred. If Min-K were a valid metric for measuring forgetting, we would expect to see a significant decrease in AUC scores after unlearning, indicating reduced ability to identify membership. Instead, both scores remain similarly poor, suggesting that Min-K fails to capture whether private information has been successfully removed from the model.

## D Evaluation

### D.1 Experimental Setup

**Model Setup.** We start with a general pre-trained base model and finetune two models: $f_{target}$ on $\mathbb{D}_F \cup \mathbb{D}_R$, and $f_{retrain}$ on $\mathbb{D}_R$ only. For each unlearning algorithm $U$, we further generate the unlearned model $f_{unlearn} = U(f_{target}, \mathbb{D}_{F_k}, \mathbb{D}_R)$.

We conduct experiments using LLaMA-3.2-3B (Dubey et al., 2024) and GPT-2-Large (Radford et al., 2019). For LLaMA-3.2-3B, we start from its publicly released checkpoint[2] then finetune on the combination of our privacy datasets Enron and MUSE News for 5 epochs. We use a cosine learning rate scheduler with an initial learning rate of $10^{-5}$ and distribute training across 2 GPUs, each processing a batch size of 2, and accumulating gradients for 32 steps before performing a backward pass. This setup effectively simulates training with a batch size of 128.

For GPT-2-Large, we start from the pre-trained model in Huggingface[3]. We then finetune on the same datasets, stopping when validation perplexity stabilizes without increasing. We use an AdamW optimizer with a batch size of 4.

**Unlearning Configuration.** Following prior work Shi et al. (2024b), for LLaMA-3.2-3B, we run all unlearning methods with a constant learning rate of $10^{-5}$ and batch size of 32. Additionally, for GPT-2-Large, we use a learning rate of $10^{-5}$ and batch size of 16. For WHP and Task Vector, we obtain the reinforced model $f_{reinforce}$ by fine-tuning $f_{target}$ on forget set for 10 epochs with the same hyperparameters. Before evaluation, we select optimal hyperparameters for each method based on utility preservation on a validation set split from retain data. For GA and NPO variants, we train for maximum 10 epochs. For WHP and Task Vector, we tune the interpolation parameter $\alpha$ in range $[0, 1]$. For NPO, we set $\beta = 0.1$. For RMU, we set the updated layers from layer 20 to layer 27, and use lay 20's activation to compute the loss function. Methods with GDR/KLR regularizers use a held-out portion of retain data ("retain1") distinct from our evaluation set ("retain2"). All experiments are conducted using 4 NVIDIA A100 GPUs with 40GB memory each. We report averaged results over 3 runs with different random seeds.

---

[2]https://huggingface.co/meta-llama/Llama-3.2-3B
[3]https://huggingface.co/openai-community/gpt2-large

Table 3: Privacy and utility measurements on Enron and News using GPT-2-Large. **U1** uses 10-shot in-context learning.

| Methods | P1. Direct Retrieval | | P2. Recovery via ICL | | P3. Recovery via Fine-tuning | | | U1. Utility Preserv. | |
| --- | --- | --- | --- | --- | --- | --- | --- | --- | --- |
| | $Rec$ on $\mathbb{D}_{F_k}$ | $Rec$ on $\mathbb{D}_{F_{uk}}$ | $Rec$ on $\mathbb{D}_{F_k}$ | $Rec$ on $\mathbb{D}_{F_{uk}}$ | $Rec$ on $\mathbb{D}_{F_k}$ | $Rec$ on $\mathbb{D}_{F_{uk}}$ | Cost | QA | QA+ICL |
| Target $f_{target}$ | 11.33% | | 39.02% | | 46.72% | | $|D| = 50, 10e$ | 7.07 | 8.70 |
| Retrain $f_{retrain}$ | 0.00% | | 35.17% | | 42.07% | | $|D| = 50, 10e$ | 7.43 | 8.75 |
| **20% Enron + News** | | | | | | | | | |
| GA | 0.00% | 0.00% | 0.00% | 0.00% | 22.68% | 19.78% | $|D| = 50, 10e$ | 0.00 | 0.00 |
| GA$_{GDR}$ | 0.00% | 0.00% | 0.12% | 3.86% | 35.85% | 33.93% | $|D| = 50, 10e$ | 8.95 | 6.56 |
| GA$_{KLR}$ | 0.00% | 0.00% | 0.12% | 1.07% | 39.51% | 37.20% | $|D| = 50, 10e$ | 8.87 | 8.38 |
| NPO | 0.00% | 0.00% | 0.00% | 0.00% | 19.15% | 16.11% | $|D| = 50, 10e$ | 1.09 | 4.13 |
| NPO$_{GDR}$ | 0.00% | 0.00% | 20.49% | 19.23% | 40.12% | 38.85% | $|D| = 50, 10e$ | 7.69 | 8.54 |
| NPO$_{KLR}$ | 0.00% | 0.00% | 29.51% | 29.67% | 38.78% | 37.60% | $|D| = 50, 10e$ | 8.53 | 9.33 |
| Task Vector | 0.00% | 0.00% | 12.68% | 25.66% | 23.95% | 28.03% | $|D| = 50, 10e$ | 7.31 | 8.66 |
| RL | 0.00% | 0.03% | 53.29% | 36.74% | 56.10% | 49.33% | $|D| = 50, 10e$ | 5.45 | 8.49 |
| RL$_{GDR}$ | 0.00% | 0.09% | 60.85% | 49.88% | 56.22% | 49.36% | $|D| = 50, 10e$ | 7.46 | 7.74 |
| RM | 4.15% | 1.22% | 49.27% | 40.54% | 45.85% | 40.14% | $|D| = 50, 10e$ | 6.21 | 6.45 |
| RM$_{GDR}$ | 6.46% | 2.08% | 51.10% | 45.65% | 45.98% | 39.19% | $|D| = 50, 10e$ | 7.53 | 8.09 |
| WHP | 0.00% | 0.00% | 13.11% | 14.06% | 40.06% | 39.86% | $|D| = 50, 10e$ | 6.48 | 7.55 |
| WHP$_{GDR}$ | 0.00% | 0.00% | 15.53% | 14.99% | 41.68% | 41.13% | $|D| = 50, 10e$ | 7.25 | 7.51 |
| IDK | 0.00% | 0.09% | 32.91% | 32.96% | 45.68% | 47.28% | $|D| = 50, 10e$ | 7.95 | 8.95 |
| IDK$_{GDR}$ | 0.00% | 0.09% | 30.91% | 31.69% | 47.82% | 46.76% | $|D| = 50, 10e$ | 7.97 | 8.48 |
| **50% Enron + News** | | | | | | | | | |
| GA | 0.00% | 0.00% | 0.00% | 0.00% | 0.24% | 0.18% | $|D| = 50, 10e$ | 0.00 | 0.00 |
| GA$_{GDR}$ | 0.00% | 0.00% | 0.00% | 0.00% | 29.34% | 29.60% | $|D| = 50, 10e$ | 9.39 | 5.88 |
| GA$_{KLR}$ | 0.00% | 0.00% | 0.00% | 0.00% | 30.12% | 31.21% | $|D| = 50, 10e$ | 9.35 | 6.66 |
| NPO | 0.00% | 0.00% | 0.00% | 0.00% | 18.73% | 18.66% | $|D| = 50, 10e$ | 1.63 | 1.31 |
| NPO$_{GDR}$ | 0.00% | 0.00% | 25.49% | 32.42% | 32.52% | 31.90% | $|D| = 50, 10e$ | 7.67 | 10.07 |
| NPO$_{KLR}$ | 0.00% | 0.00% | 23.78% | 26.91% | 34.47% | 35.27% | $|D| = 50, 10e$ | 7.82 | 9.46 |
| Task Vector | 0.00% | 0.00% | 14.43% | 15.53% | 21.92% | 25.44% | $|D| = 50, 10e$ | 8.93 | 8.69 |
| RL | 0.00% | 0.00% | 64.02% | 59.68% | 58.34% | 54.08% | $|D| = 50, 10e$ | 7.15 | 8.19 |
| RL$_{GDR}$ | 0.00% | 0.00% | 63.54% | 56.00% | 58.83% | 55.20% | $|D| = 50, 10e$ | 7.58 | 8.84 |
| RM | 2.30% | 0.49% | 45.53% | 40.99% | 46.06% | 39.86% | $|D| = 50, 10e$ | 6.48 | 7.55 |
| RM$_{GDR}$ | 3.81% | 0.93% | 53.11% | 44.06% | 47.68% | 41.13% | $|D| = 50, 10e$ | 7.25 | 7.51 |
| WHP | 0.00% | 0.00% | 18.97% | 19.99% | 33.12% | 35.82% | $|D| = 50, 10e$ | 6.35 | 7.38 |
| WHP$_{GDR}$ | 0.00% | 0.00% | 4.29% | 5.27% | 38.29% | 41.13% | $|D| = 50, 10e$ | 7.92 | 8.00 |
| IDK | 0.00% | 0.00% | 32.91% | 32.96% | 44.58% | 44.80% | $|D| = 50, 10e$ | 7.95 | 8.85 |
| IDK$_{GDR}$ | 0.00% | 0.00% | 30.91% | 31.69% | 47.82% | 46.21% | $|D| = 50, 10e$ | 7.81 | 8.94 |

## D.2 EXPERIMENTAL RESULTS ON GPT-2

Our experiments on GPT-2-Large Table 3 confirm the generalizability of findings observed in LLaMA-3.2, demonstrating that the vulnerability to active attacks is not architecture-specific. While GPT-2 shows more severe utility degradation compared to LLaMA-3.2's robustness, the fundamental patterns remain consistent: training pipeline methods outperform data manipulation approaches, and substantial P1-P3 gaps persist across most unlearning methods, indicating shallow forgetting as a universal limitation.

