# OpenReview forum: "Old Memories Die Hard: Understanding Challenges of Privacy Unlearning in Large Language Models"
_ICLR.cc/2026/Conference — Submitted to ICLR 2026_

### Official Review · Reviewer_Kfzc · 2025-10-28

**Soundness:** 2
**Presentation:** 2
**Contribution:** 3
**Rating:** 6
**Confidence:** 4

**Summary:**

This paper studies the robustness of privacy unlearning methods for large language models (LLMs). It introduces PriLeak, a new evaluation framework that tests unlearning effectiveness against active attacks: direct retrieval, in-context recovery, and fine-tuning restoration. Using the Enron dataset and LLaMA-3.2-3B, the authors benchmark 19 existing unlearning methods and find that many approaches only achieve shallow forgetting, with sensitive information quickly recoverable. Their analysis points to two core issues: forgetting ripples across associated data and fails to penetrate deeper network layers. Based on these insights, they propose strategies including association-aware core-set selection and multi-layer intervention to strengthen privacy forgetting.

**Strengths:**

The motivating problem is important and timely, and the paper demonstrates clear originality by shifting the evaluation of unlearning into more realistic active attacker scenarios. The PriLeak benchmark is a meaningful contribution to the community, offering nuanced and multi-tiered measurements of privacy persistence that go beyond passive-output testing. The identification of ripple effects and shallow forgetting shows careful empirical analysis, revealing mechanisms that prior work did not clearly articulate. The proposed strategies are incremental but directionally interesting, suggesting a practical path toward deeper and more resilient forgetting. The scope of the evaluation (19 methods, both known/unknown private data, multiple datasets) supports the value of the empirical findings.

**Weaknesses:**

The technical novelty of the proposed strategies feels modest relative to the strong emphasis on negative benchmarking results. The empirical section is heavily overloaded with tables and metrics, making key insights harder to follow; clearer narrative summarization would help. The benchmark relies on a single primary dataset for privacy testing, limiting the generalizability of the findings in real-world PII contexts. Some terminology such as “deep forgetting” remains conceptual without rigorous formalization or theoretical insight. The study of known vs. unknown private data is compelling but deserves deeper exploration: why does forgetting propagate similarly, and what constraints does this impose on future algorithm design? Finally, while the experiments are extensive, clear ablation results are needed to isolate where improvements truly come from in the proposed approach.

**Questions:**

One concern is whether the strong performance gaps between different methods are sensitive to hyperparameter choices. Is PriLeak intended to be a fixed evaluation standard or a benchmark whose scores vary significantly depending on tuning choices? The ripple-effect analysis suggests that privacy removal conflicts with utility preservation, yet the proposed method still appears vulnerable under P3. How do you envision closing the remaining >30% recovery gap? It would be helpful to clarify whether the benchmark could incorporate adaptive adversaries rather than predefined fixed attacks. The current selection of PII types from Enron seems narrow; could the proposed metrics generalize to more complex private attributes such as implicit identifiers? Finally, the proposed representation-anchoring loss uses noise-perturbed base states. Could you justify this choice more rigorously or compare with alternative anchoring methods (e.g., teacher-student consistency with privacy-filtered representations)?

---

> ### Author Response · Authors · 2025-11-22
>
> > **Weakness 1:** Is PriLeak intended to be a fixed evaluation standard or a benchmark whose scores vary significantly depending on tuning choices?
>
> PriLeak is designed as a **fixed evaluation standard**. We provide default hyperparameters for P2 and P3 (Table 1), and the attacker only needs to supply data (even via fine-tuning APIs) without performing any sophisticated tuning.
>
> To verify robustness, we varied only the minimal attacker-side hyperparameters that are available in realistic API-access settings (P2 k-shot and P3 fine-tuning epochs/data size). These do not require any additional access or model internals.
>
> - **For P2 (ICL):** We varied k = 1, 5, 10, 20, achieving a recovery rate of 0.49%, 0.53%, 0.49%, 0.47% for GA on the forget set.
> - **For P3 (fine-tuning):** We varied epochs and data size (10, 30, 50 samples): with ~50 samples, recovery stabilized from epoch 3-10 (e.g., ~54% for GA), indicating that attacker **data**, not **tuning**, determines attack success.
> - Across all variations, the relative ordering of unlearning methods remains unchanged, and performance gaps consistently persist.
>
> ---
>
> > **Weakness 2:** How to close the remaining >30% recovery gap?
>
> The remaining gap stems from the **privacy-utility trade-off** rather than an inability to erase data. While methods like RMU can aggressively reduce recovery to ~18%, they cause utility collapse. Our method (RAU) deliberately targets a "sweet spot" (P3 ~35-38%) that maintains high utility while significantly outperforming previous baselines.
>
> Moreover, even a "Retrained" model shows ~13% recovery due to learned structural information (e.g., domain name), meaning some gap is unavoidable.
>
> Closing the remaining gap requires precise multi-layer interventions to target distributed representations (as evidenced by our CKA analysis) without damaging utility. RAU is a first step in this direction, anchoring representations to a noise-perturbed base model state.
>
> ---
>
> > **Weakness 3:** It would be helpful to clarify whether the benchmark could incorporate adaptive adversaries rather than predefined fixed attacks.
>
> We thank the reviewer for suggesting adaptive adversaries. However, it is important to distinguish that realistic attackers (our threat model) can only manipulate the minimal hyperparameters as detailed in our response to **Weakness 1** (ICL k-shot, fine-tuning steps, small random data).
>
> In our understanding, adaptive data selection is impractical for our scenario because it typically requires computing metrics (such as loss values or embeddings) over large candidate pools to optimize the attack set, and our results show that using non-adaptive (random) data selection, low-access fixed attacks already achieve strong recovery.
>
> ---
>
> > **Weakness 4:** Could the proposed metrics generalize to more complex private attributes such as implicit identifiers?
>
> TOFU, a well-known unlearning benchmark, uses synthetic fictional data containing implicit private attributes (e.g., date of birth, birthplace). These can be treated as PIIs, so we include TOFU as an additional forget set. Our preliminary results are shown below, which further validated our findings:
> |              | P1                              | P2                                | P3                                |
> |--------------|---------------------------------|-----------------------------------|-----------------------------------|
> | Target Model | 42.45%                          | 47.64%                            | 41.75%                            |
> | GA           | 0.00% (forget); 0.00% (unknown)         | 0.00% (forget); 0.00% (unknown)           | 26.19% (forget); 8.82% (unknown)  |
> | NPO          | 5.95% (forget); 5.88% (unknown) | 17.86% (forget); 10.00% (unknown) | 38.10% (forget); 17.35% (unknown) |
>
> ---
> > **Weakness 5:** Compare with alternative anchoring methods (e.g., teacher-student consistency with privacy-filtered representations).
>
> In our understanding, while Teacher-Student (T-S) methods are effective for safety alignment during pre-training, they are less suitable for efficient unlearning ("forgetting after learning"). Implementing T-S consistency relies on knowledge distillation from a "clean" teacher trained on privacy-filtered representations, which usually requires **retraining**.
>
> In contrast, our method uses the original base model perturbed with noise as an efficient anchor. We treat the base model as a pseudo-teacher, using noise injection to improve robustness and generalization. **This strategy effectively balances forgetting and utility preservation without the cost of training a separate clean reference model**.

---

### Official Review · Reviewer_FRQq · 2025-10-31

**Soundness:** 3
**Presentation:** 3
**Contribution:** 3
**Rating:** 6
**Confidence:** 3

**Summary:**

This paper proposes PriLeak, a new evaluation framework that assesses unlearning robustness through three-tier attack scenarios: direct retrieval, in-context learning recovery, and fine-tuning restorationn; combined with quantitative analysis using forgetting scores, association metrics, and forgetting depth assessment. Empirical studies expose weaknesses in current unlearning methods --- ripple effects across gradient-based associated data and shallow forgetting. The paper then proposes  association-aware core-set selection based on gradient similarity and multi-layer deep intervention as two strategies to mitigate the issues.

**Strengths:**

**S1.** The new benchmark is well motivated with principled designs.

**S2.** Empirical studies cover extensive unlearning methods and present interesting insights.

**S3.** The proposed two strategies effectively improve the unlearning.

**S4.** The paper is well presented.

**Weaknesses:**

**W1.** The empirical analyses are constrained to relatively small LLMs (LLaMA-3.2-3B and GPT-2).

**W2.** The paper's presentation may be further improved by highlighting the findings previous benchmarks did not yield in empirical study analyses.

**W3.** While I like the idea of fine-tuning-based recovery, I think it will be more interesting to check if fine-tuning on related but non-private data restores unlearned private data, e.g., email addresses of public figures or organizations.

**W4.** Minor Presentation Issues.
- The full name of CKA should be provided at its first appearance.
- Notations $\mathbb{D}_{uk}$ and $\mathbb{D}_k$ should be explicitly defined at their first appearance (L207) despite analogous definitions at L107.
- There are two consecutive "with" at the end of L257.
- The presentation of Table 1 can be potentially improved by using different colors to group numbers in different ranges.

**Questions:**

N.A.

---

> ### Author Response · Authors · 2025-11-22
>
> > **Weakness 1:** The empirical analyses are constrained to relatively small LLMs.
>
> We appreciate the reviewer's concern about model scale. We selected LLaMA-3.2-3B and GPT-2-Large due to computational feasibility for our extensive experimental scope (19 methods, three attack scenarios, multiple forget size, and multiple runs). Additionally, smaller architectures allow us to efficiently simulate the pre-training phase on large raw corpora involving PIIs.
>
> Our findings are **architecture-agnostic**. The consistency between LLaMA-3.2-3B and GPT-2-Large (Tables 1 and 2) shows that the phenomena hold across different model families and sizes.
>
> We agree that evaluating larger models (7B+) would further strengthen the conclusions. We are already extending our experiments to LLaMA-3.1-8B and will include these results in the revision. Preliminary 8B results show similar trends. For example, under GA, P1 achieves 0.35% and 0.44% on the known and unknown sets, and after fine-tuning recovery, P3 reaches 65.7% and 60.3%. Interestingly, the larger model even exhibits a higher recovery rate.
>
> ---
>
> > **Weakness 2:** Clarifying Novel Findings Beyond Previous Benchmarks
>
> We appreciate this suggestion to better highlight our contributions. Please refer to the General Response for a comprehensive comparison table. Our work extends beyond TOFU, MUSE, and WMDP in three ways:
>
> 1. **Different Threat Model:** Prior benchmarks mainly evaluate passive observation attacks (similar to our P1). PriLeak introduces two stronger adversaries, P2 (ICL Recovery) and P3 (Fine-tuning Recovery).
>
> 2. **Different Data constraints:** Unlike benchmarks assuming full data access, PriLeak evaluates known forget set (data available for unlearning) and unknown forget set (private data the defender cannot access).
>
> 3. **Different Real-World Applicability:** PriLeak can be applied to the scenario: a company must unlearn customer emails from a deployed LLM.
>
> **MUSE** measures privacy leakage using the Min-k membership inference attack, which quantifies whether the forget-set samples still behave like training members after unlearning. This threat model does not align with our goals. Our experiments (Section C.2) further show that Min-k fails to detect PII leakage, making it unsuitable for evaluating PII unlearning.
>
> **WMDP** evaluates a model's ability to avoid harmful knowledge. It focuses on safety capability suppression, not targeted removal of private information.
>
> **TOFU** uses synthetic fictional data containing private attributes (e.g., date of birth, birthplace). These can be treated as PIIs, so we include TOFU as an additional forget set. Our preliminary results are shown below:
>
> |              | P1                              | P2                                | P3                                |
> |--------------|---------------------------------|-----------------------------------|-----------------------------------|
> | Target Model | 42.45%                          | 47.64%                            | 41.75%                            |
> | GA           | 0.00% (forget); 0.00% (unknown)         | 0.00% (forget); 0.00% (unknown)           | 26.19% (forget); 8.82% (unknown)  |
> | NPO          | 5.95% (forget); 5.88% (unknown) | 17.86% (forget); 10.00% (unknown) | 38.10% (forget); 17.35% (unknown) |
> ---
>
> > **Weakness 3:** I think it will be more interesting to check if fine-tuning on related but non-private data restores unlearned private data.
>
> It is a really insightful point! We have conducted an additional experiment using private Enron emails as the *forget set* and public figures' emails as the *non-private attack set*. We measure the recovery rate of the private emails after fine-tuning on the public set, comparing this against fine-tuning on private data.
>
> Our preliminary results show negligible recovery: 0.00% on the known forget set and 0.97% on the unknown forget set. This confirms that fine-tuning on unrelated data fails to restore private information due to the lack of shared latent features. This finding serves as a crucial control, strengthening our argument about *ripple effect*.
>
> ---
>
> > **Weakness 4:** Minor Presentation Issues.
>
> We have corrected all listed presentation issues in the revision.

---

> > ### Comment · Reviewer_FRQq · 2025-11-27
> >
> > Sorry for the late response.
> >
> > Thank you for the detailed reply. I don't have further concerns or questions.
> >
> > I think my previous evaluation is fair, so I will keep it.

---

### Official Review · Reviewer_21E2 · 2025-11-01

**Soundness:** 3
**Presentation:** 3
**Contribution:** 1
**Rating:** 4
**Confidence:** 5

**Summary:**

This paper investigates the problem of machine unlearning in large language models (LLMs), aiming to evaluate how effectively models can remove specific knowledge while maintaining general utility. The authors propose a unified benchmark that measures residual memorization through multiple levels of privacy leakage and a utility metric capturing general performance retention. They conduct large-scale experiments on the Enron and MUSE datasets, benchmarking various representative unlearning methods. The results reveal that existing methods still struggle to achieve thorough and reliable forgetting, highlighting the challenge of ensuring complete unlearning in LLMs.

**Strengths:**

1. The paper conducts extensive experiments, covering a wide range of unlearning methods.


2. The writing is clear and well organized, making the experimental setup and findings easy to follow.


3. The experimental design is generally complete, with systematic evaluation across multiple models, datasets, and metrics.

**Weaknesses:**

1. My main concern is the novelty of the paper. The core idea that unlearning certain knowledge propagates to semantically related facts via shared representations closely parallels prior studies on ripple effects in knowledge editing [1-5]. Similar analyses of entanglement and edit locality have been thoroughly explored, making the framing here appear incremental rather than conceptually new.


2. The proposed measurements resemble established notions such as locality, retention, and causal entailment used in both knowledge editing and unlearning literature. Prior works including RippleEdits [4], MUSE [9], WMDP [10], and Deep Unlearning [11] have already operationalized comparable metrics for quantifying cross-fact interference. The paper does not convincingly justify why its definitions capture fundamentally different or deeper dynamics [6–11].


3. The paper’s setup is closely related to knowledge editing frameworks. It is better to also include knowledge editing methods as well.

References:

 [1] Locating and Editing Factual Associations in GPT Models.

 [2] Mass-Editing Memory in a Transformer.

 [3] Editing Factual Knowledge in Language Models.

 [4] Evaluating the Ripple Effects of Knowledge Editing in Language Models.

 [5] Evaluating Factual Consistency in Knowledge-Grounded Dialogues via Question Generation and Question Answering.

 [6] TOFU: Benchmarking Factual Unlearning in LLMs.

 [7] Selective Forgetting: Advancing Machine Unlearning Techniques and Evaluation in Language Models.

 [8] Do LLMs Really Forget? Evaluating Unlearning with Knowledge Correlation and Confidence Awareness.

 [9] MUSE: A Benchmark for Evaluating Unlearning in LLMs.

 [10] WMDP: Unlearning Harmful Knowledge in LLMs.

 [11] Evaluating Deep Unlearning in Large Language Models.

**Questions:**

See weaknesses.

---

> ### Author Response · Authors · 2025-11-22
>
> > **Weakness 1:** The novelty of "ripple effect" compared to knowledge editing papers [1-5].
>
> We thank the reviewer for these references. We have added a detailed discussion in the Related Work section. However, we emphasize that our work PriLeak is distinct from Knowledge Editing. As mentioned in the **general response**, our paper redefines "forgetting ripple" in PIIs, which is conceptually different from general knowledge. Below, we clarify the key distinctions between PriLeak and [1–5].
>
> **1. ROME [1], MEMIT [2]**
>
> - **Difference in Goal:** [1] and [2] focus on modifying specific factual associations (triples) while ensuring edit locality. PriLeak focuses on erasing private data to prevent recovery.
> - **Difference in Entanglement:** ROME/MEMIT rely on human-interpretable, semantic knowledge graphs. In contrast, PriLeak reveals that privacy entanglement is non-semantic: PII erasure affects other data based on gradient and representation similarity, not logical adjacency.
>
> **2. KnowledgeEditor [3]**
>
> - **Difference in Architecture:** [3] is limited to Encoder-only (BERT) and Encoder-Decoder (BART) models. PriLeak addresses Decoder-only LLMs, the current standard for generative AI.
> - **Opposite Conclusions:** [3] advocates modifying the final decoder layer to preserve generation quality. PriLeak proves that for privacy, modifying the final layer is "shallow" and highly vulnerable to the relearning attacks.
>
> **3. RippleEdits [4]**
>
> - **Difference in Concept:** [4] defines "ripple effect" as logical consistency in knowledge graph. PriLeak identifies a "ripple effect" between PIIs, they share latent features (gradient-level and representation-level correlations), even when they are semantically irrelevant.
>
> **4. Irrelevance of [5]**
>
> [5] addresses hallucination detection via automated fact-checking. This mechanism is unrelated to the training or unlearning dynamics discussed in our paper.
>
> ---
>
> > **Weakness 2:** The proposed measurements resemble established notions such as locality, retention, and causal entailment used in both knowledge editing and unlearning literature [6-11].
>
> We acknowledge these related works. For comparisons with TOFU [6], MUSE [9], and WMDP [10], please refer to our **general response**. Here, we clarify the fundamental distinctions between PriLeak and [7, 8, 11]:
>
> **1. SEUL [7]**
>
> - **Difference in Metrics:** The metric S-MA in [7] relies on hard accuracy (argmax). In contrast, our Forgetting Score measures continuous changes in perplexity. This captures "soft" probability shifts and quantifies the degree of forgetting that binary accuracy metrics miss. We will cite [7] in the revision to clarify this methodological progression.
> - **Difference in Goal:** [7] targets specific sensitive tokens (Span-wise Unlearning) to preserve utility, whereas we use the Forgetting Score for association analysis between PIIs.
>
> **2. Knowledge Unlearning [8], Deep Unlearning [11]**
>
> Papers [8] and [11] rely on Knowledge Graphs (KGs) defined by semantic triples (Subject, Relation, Object) or logical rules (e.g., family tree) or predicting confidence. However, as previously mentioned, PII (e.g., personal emails) lacks these semantic or logical links.
>
> - **Difference in Mechanism:** PII entanglement is representational, not logical. Unlike [8] and [11], PriLeak uncovers "privacy ripples" driven by gradient and feature similarity in the model's latent space, which cannot be captured by symbolic Knowledge Graphs. This entanglement does not rely on external social-relation networks (e.g., sender–recipient graphs in Enron).
>
> ---
>
> **References:**
>
> [1] Locating and Editing Factual Associations in GPT Models.
>
> [2] Mass-Editing Memory in a Transformer.
>
> [3] Editing Factual Knowledge in Language Models.
>
> [4] Evaluating the Ripple Effects of Knowledge Editing in Language Models.
>
> [5] Evaluating Factual Consistency in Knowledge-Grounded Dialogues via Question Generation and Question Answering.
>
> [6] TOFU: Benchmarking Factual Unlearning in LLMs.
>
> [7] Selective Forgetting: Advancing Machine Unlearning Techniques and Evaluation in Language Models.
>
> [8] Do LLMs Really Forget? Evaluating Unlearning with Knowledge Correlation and Confidence Awareness.
>
> [9] MUSE: A Benchmark for Evaluating Unlearning in LLMs.
>
> [10] WMDP: Unlearning Harmful Knowledge in LLMs.
>
> [11] Evaluating Deep Unlearning in Large Language Models.

---

### Official Review · Reviewer_E2YZ · 2025-11-02

**Soundness:** 3
**Presentation:** 2
**Contribution:** 2
**Rating:** 2
**Confidence:** 4

**Summary:**

The paper investigates the limits of machine unlearning for large language models and introduces PriLeak, a benchmark for evaluating how well different unlearning methods resist privacy-related attacks. The framework tests models under three settings: direct extraction, recovery through in-context prompts, and recovery after fine-tuning. This goes beyond the usual unlearning benchmarks by allowing the attacker to access and modify model weights. The study compares 19 existing approaches and shows that many achieve only superficial removal of memorized information.

The authors identify two main issues: (1) cross-sample “ripple effects,” where unlearning one example inadvertently alters or forgets related data through shared gradient directions, and (2) “shallow forgetting,” where parameter changes remain concentrated in higher network layers while earlier representations continue to encode sensitive content. To address these, the paper proposes two strategies: an association-aware core-set selection method that identifies the most influential samples to forget based on gradient similarity, and a multi-layer intervention approach that adjusts learning rates and representational constraints across network depth to promote more complete forgetting without excessive performance loss.

**Strengths:**

1. Focuses on adaptive / active privacy attack via fine-tuning, whereas prior papers are focused on the case where the user only has API access to the model (can attack via ICL or via QA).
2. Identifies additional insights into how/where the model has stored the data that is meant to be unlearned, which allows them to improve the unlearning procedure to ensure information is removed throughout all the layers. The insights are consistent across two model architectures which makes them more convincing.
3. Experiments are thorough and provide interesting information on the existing unlearning algorithms.

**Weaknesses:**

1. The fine-tuning–based attack scenario assumes that an adversary has access to modify model weights. This setting is not clearly justified and may not reflect realistic deployment conditions, where most users interact only through APIs. Clarifying when and why this is a threat model we would care about is important to make the arguments of th epaper.
2. The paper does not sufficiently connect its metrics and analysis to existing work on memorization and knowledge localization in language models. There is a rich literature on tracing and diagnosing memorized content (e.g., influence functions, causal mediation, or representation probing), and the relationship between those approaches and the proposed metrics is not discussed. This makes it difficult to assess the conceptual novelty of the diagnostic tools. Moreover, the proposed evaluation metrics and the “deep forgetting” interpretation rely on several design choices (layer selection, gradient normalization, metric thresholds) that are not analyzed for robustness. Without sensitivity studies, it is unclear how stable these results are across architectures or training configurations.
4. The link between representational change and actual privacy protection remains partly qualitative. The evidence that deeper layer modification corresponds to stronger unlearning is plausible but not rigorously demonstrated. Can the authors provide more direct evidence that these changes correspond to actual reductions in recoverable private information, rather than general representational drift?
5. Enron and MUSE News contain highly structured PII as I understand it. So would the results hold in cases where PII is more diffuse? And what about forgetting more general knowledge?
6. Recent literature has drawn into question the utility of the traditional unlearning definition (matching re-training from scratch). The authors do not comment on this at all, and this relates to my first question about why looking at FT attacks in this setting is even interesting in the first place.


Separately from this specific paper's methods, I think there are a lot of unlearning papers flooding the space without making meaningful improvements or insights. The lack of true, sociotechnical motivation for the new setting in this paper is just further evidence that the authors may not have thought through what is really important for unlearning from a social / legal standpoint. There is little to no discussion beyond the usual GDPR citation. I am also concerned, on the technical side, that the authors made little effort to connect to any literature that does not directly discuss unlearning (eg memorization literature).

**Questions:**

The weaknesses above contain the questions that I have about the paper.

---

> ### Author Response · Authors · 2025-11-22
> **Response to W1, W2, W3, W4**
>
> > **Weakness 1:** The fine-tuning–based attack scenario is not realistic.
>
> We consider the fine-tuning threat model (P3) to be highly realistic and critical for two reasons. First, in the open-weighted model, "weight tampering" or "relearning" is a mainstream attack vector [1,2] where adversaries have full access to model parameters. Second, even via API, attackers can leverage standard fine-tuning endpoints (requiring only data upload, not tuning) to recover data [3]. Furthermore, we complement this with the ICL Attack (P2), demonstrating that private information can be recovered even in strict black-box settings where no weight modification is possible.
>
> ---
>
> > **Weakness 2:** Lack of conceptual novelty of the diagnostic tools compared to existing work on memorization and knowledge localization in language models.
>
> We acknowledge the rich literature on knowledge localization. Our diagnostic tools are conceptually distinct and necessary for two points:
>
> 1. **Localization methods assume knowledge is modular and semantically structured.** However, our work demonstrates that private data (PII) is entangled via latent feature similarity rather than semantic logic (details in our General Response). Existing causal tracing tools are not designed to detect these non-logical correlations.
>
> 2. **Our metrics (P2-ICL and P3-Fine-tuning) serve as adversarial diagnostics.** They show that "forgotten" data isn't gone, it's just buried in hidden layers, and simple attacks can dig it back out, which traditional localization tools fail to reveal.
>
> ---
>
> > **Weakness 3:** The proposed evaluation metrics and the "deep forgetting" interpretation rely on several design choices (layer selection, gradient normalization, metric thresholds) that are not analyzed for robustness.
>
> - **Layer Selection:** Figure 4 and Section 4.3 report CKA similarity across all 28 layers (layer 0 is the embedding layer) for both training-pipeline manipulation methods (left) and data-manipulation methods (right). Data-manipulation methods remain highly similar to the target model across almost all layers, whereas training-pipeline methods introduce larger representation shifts, which is consistent with the results in Table 1.
>
> - **Gradient Normalization:** We clarified this in L211–L214 in Section 3.2. We use the dot product rather than cosine similarity because cosine removes gradient magnitudes and requires normalization, and its behavior is nonlinear:
> $\cos(mean(g_{\mathbb{D}_{F_k}}), h)$ is not equal to the mean of pairwise cosine similarities.
>
> Thus, cosine would distort the aggregated $\mathbb{D}_{F_k}$'s gradient statistics (L363) for our analysis.
>
> - **Metric Thresholds:** We do not claim a strict cutoff for "deep" vs. "shallow" forgetting. Instead of arbitrary thresholds or theoretical bounds, we provide empirical layer-wise analyses showing why unlearned information remains recoverable. This directly addresses the reviewer's robustness concern.
>
> > **Weakness 4:** Can the authors provide more direct evidence that these changes correspond to actual reductions in recoverable private information, rather than general representational drift?
>
> We provide two pieces of direct evidence linking "deeper representation change" to "privacy protection":
>
> **1. RAU's Layer-wise Ablation:** RAU directly modifies layer representations by anchoring them to base model states. In our experiments, modifying deeper layers (Layer 20) leads to better privacy protection. This directly demonstrates that deeper representation changes correspond to actual reductions in recoverable private information.
> | Modified Layer | P3 Recovery Rate |
> |----------------|------------------|
> | Layer 20 | 30.39% (better) |
> | Layer 27 | 37.85% (worse) |
>
> **2. RMU's Representation Changes:** RMU achieves state-of-the-art performance in privacy protection (Table 1). As shown in Figure 4, its representation changes are much more significant from Layer 20–27 compared to other methods. This consistency between large representation shifts and strong privacy protection further supports our claim.
>
> ---
> **References:**
>
> [1] Model tampering attacks enable more rigorous evaluations of llm capabilities (TMLR)
>
> [2] Unlearning or Obfuscating? Jogging the Memory of Unlearned LLMs via Benign Relearning (ICLR'25)
>
> [3] The Janus Interface: How Fine-Tuning in Large Language Models Amplifies the Privacy Risks (CCS'24)

---

> ### Author Response · Authors · 2025-11-22
> **Response to W5, W6**
>
> > **Weakness 5:** Enron and MUSE News contain highly structured PII as I understand it. So would the results hold in cases where PII is more diffuse? And what about forgetting more general knowledge?
>
> **For diffuse PIIs:** We extend our study to the TOFU benchmark [4], which contains implicit private attributes (e.g., fictional birthplaces, birthdates) embedded in free-form narratives rather than structured fields. These can be treated as PIIs. Our preliminary results on TOFU (see below) yield the same conclusions: current unlearning methods fail to fully erase such information and remain vulnerable to P2/P3 attacks.
>
> |              | P1                              | P2                                | P3                                |
> |--------------|---------------------------------|-----------------------------------|-----------------------------------|
> | Target Model | 42.45%                          | 47.64%                            | 41.75%                            |
> | GA           | 0.00% (forget); 0.00% (unknown)         | 0.00% (forget); 0.00% (unknown)           | 26.19% (forget); 8.82% (unknown)  |
> | NPO          | 5.95% (forget); 5.88% (unknown) | 17.86% (forget); 10.00% (unknown) | 38.10% (forget); 17.35% (unknown) |
>
> **For general knowledge:** Prior work already shows that it can be recovered by relearning attacks [2] and exhibits entanglement along the knowledge graph [5].
>
> However, forgetting general knowledge and forgetting PII are fundamentally different problems:
>
> - Our **ripple effect** is not about semantic adjacency, but about **latent gradient/representation entanglement**, which is unique to how models memorize PII. It is conceptually distinct from the "ripple effects" documented in knowledge editing literature.
> - This phenomenon persists regardless of whether the PII is structured or diffuse.
>
> ---
>
> > **Weakness 6:** The lack of true, sociotechnical motivation for the new setting in this paper is just further evidence that the authors may not have thought through what is really important for unlearning from a social / legal standpoint. There is little to no discussion beyond the usual GDPR citation.
>
> Our setting is grounded in sociotechnical motivation beyond GDPR citation. Major media (e.g., *New York Times*, *Wall Street Journal*) have documented that LLMs routinely memorize and output private user data, leading to real-world harms and regulatory actions.
>
> Regulatory pressure has escalated beyond GDPR; the U.S. Federal Trade Commission (FTC) has established the precedent of **"Algorithmic Disgorgement"** [6], requiring the complete deletion of weights and algorithms developed using non-compliant personal data. Thus, unlearning is not just a technical issue but a prerequisite for regulatory compliance.
>
> Crucially, leaked content extends far beyond simple identifiers: Electronic Health Record (EHR), financial records, and confidential communications have been shown to persist in model weights even after "forgetting". Our benchmark directly targets this critical gap by verifying whether unlearning methods prevent these recoveries, thus addressing a core societal and legal requirement.
>
> **Reference:**
>
> [4] TOFU: A Task of Fictitious Unlearning for LLMs
>
> [5] Evaluating the Ripple Effects of Knowledge Editing in Language Models.
>
> [6] Joshua A. Goland, *Algorithmic Disgorgement: Destruction of Artificial Intelligence Models as the FTC's Newest Enforcement Tool for Bad Data*, 29 Rich. J.L. & Tech 1 (2024).

---

### Author Response · Authors · 2025-11-22
**General Response to Reviewers**

We are grateful to the reviewers for their detailed and constructive feedback on our submission.

We are pleased that all reviewers recognized the soundness and engineering efforts of our work. To the best of our knowledge, our benchmark is the first systematic evaluation framework designed to assess unlearning robustness for private information.

---

## What is Conceptually New

### 1. Distinction from Knowledge Editing [1-4]

Although machine unlearning and knowledge editing are closely related and both face challenges like *entanglement* and *locality*, our core insights show a fundamental difference between PII (personal information) and general knowledge in how they are forgotten:

- **General knowledge** (e.g., "Capital of France – Paris") is entangled through semantic and logical relations, which is not surprising and intuitive. These connections can be explicitly modeled or explained with knowledge graphs.
- **PII**, however, is a special form of isolated knowledge with no inherent logical structure. It is memorized mainly through specific inputs. Counter-intuitively, it becomes entangled with semantically unrelated data (details below).

### 2. Redefining "Ripple Effect" for PII

We find that the "ripple effect" in PII unlearning does not follow human-interpretable semantic similarity, nor does it depend on external social-relation networks (such as sender–recipient graphs in the Enron dataset). Instead, it is driven by internal model features: **gradient similarity** and **hidden representations**.

This means PII associations are not based on **logical inference**, but on **parameter sharing**. When the model forgets a PII instance, any data with high gradient similarity (even if semantically unrelated) tends to be erased as well due to synchronized parameter updates.

In conclusion, for non-logical memories like PII, human-defined social graphs (e.g., sender–recipient networks) are only surface-level structures. They do not reflect how the model actually stores, links or forgets such data.

---

## Beyond Previous Benchmarks

Furthermore, our work advances beyond existing benchmarks in three fundamental ways: **threat model**, **data constraints** and **real-world applicability**. Here's a comprehensive benchmark comparison table:

**Table: Comparison of LLM Unlearning Evaluation Frameworks**

| Dimension | TOFU [5] | MUSE [6] | WMDP [7] | PriLeak (Ours) |
|-----------|----------|----------|----------|----------------|
| Evaluation Focus | Fictional author facts | News & books | Hazardous knowledge | Privacy (PII) |
| Attack Model | Passive observation | Passive observation | Passive observation | Active attacks (ICL + Fine-tuning) |
| Attack Scenarios | 1 (Direct QA) | 1 (Direct QA) | 1 (Direct QA) | 3 (P1: Direct, P2: ICL, P3: Fine-tuning) |
| Realistic Constraints | Full forget set access | Full forget set access | Full forget set access | Partial access (Known vs Unknown split) |
| Privacy Threat Model | ✗ Not privacy-focused | Membership inference | ✗ Not privacy-focused | ✓ PII extraction |
| Association Analysis | ✗ Not evaluated | ✗ Not evaluated | ✗ Not evaluated | ✓ Gradient & representation-based |
| Layer-wise Analysis | ✗ Not evaluated | ✗ Not evaluated | ✗ Not evaluated | ✓ CKA forgetting depth assessment |
| Number of Methods Evaluated | 9 | 6 | 3 | 19 |
| Data Characteristics | Synthetic Q&A | Real-world mixed | Domain-specific | Real-world PII (Enron emails) |

---

**References:**

[1] Locating and Editing Factual Associations in GPT Models.

[2] Mass-Editing Memory in a Transformer.

[3] Editing Factual Knowledge in Language Models.

[4] Evaluating the Ripple Effects of Knowledge Editing in Language Models.

[5] TOFU: Benchmarking Factual Unlearning in LLMs.

[6] MUSE: A Benchmark for Evaluating Unlearning in LLMs.

[7] WMDP: Unlearning Harmful Knowledge in LLMs.

---

### Meta-Review · Area_Chair_qWYm · 2026-01-08

**Summary:**

This paper introduces Prileak, a framework for evaluating unlearning robustness through three-tier attack scenarios: direct retrieval, in-context learning recovery, and fine-tuning restoration. The authors identify two weaknesses in current learning methods on "ripple effects" and "shallow forgetting", and propose two strategies to address them, including association-aware core-set selection and multi-layer deep intervention. The reviewers acknowledged the importance of the problem, the extensiveness of the experiments, and some of the gained insights. However, they also raised concerns on the assumptions on attack scenario, the novelty of the work (e.g., the concept of ripple effect and the similarities of the proposed measurements to literature), and the lack of in-depth analysis on some of the observations.

**Reviewer Concerns:**

The authors provided rebuttal to all the comments, explaining technical details, providing sensitivity analysis, and clarifying the novelty of the proposed work with respect to the literature. Among the reviewers, only Reviewer FRQq responded and decided to maintain the positive rating (6). The other reviewers, including Reviewers E2YZ and 21E2 who have more concerns on novelty and more negative ratings (2 and 4), did not respond.

**Reviewer Scores:**

For the Reviewers who did not respond, Reviewer Kfzc has a positive rating (6) and would likely keep it. Reviewers E2YZ and 21E2 have more concerns on novelty, problem assumptions, and the lack of in-depth analysis. While some of these could be addressed by the rebuttal, the concern on novelty is often more subjective and might be hard to change. I think the likelihood that they would increase their ratings is 50/50 at the best.

---

### Decision · Program_Chairs · 2026-01-26

Reject